# Predominant regional biophysical cooling from recent land cover changes in Europe

Bo Huang [1,5], Xiangping Hu [1,5], Geir-Arne Fuglstad [2], Xu Zhou [3], Wenwu Zhao[4] &
Francesco Cherubini[1✉]

Around 70 Mha of land cover changes (LCCs) occurred in Europe from 1992 to 2015. Despite LCCs being an important driver of regional climate variations, their temperature effects at a continental scale have not yet been assessed. Here, we integrate maps of historical LCCs with a regional climate model to investigate air temperature and humidity effects. We find an average temperature change of −0.12 ± 0.20 °C, with widespread cooling (up to −1.0 °C) in western and central Europe in summer and spring. At continental scale, the mean cooling is mainly correlated with agriculture abandonment (cropland-to-forest transitions), but a new approach based on ridge-regression decomposing the temperature change to the individual land transitions shows opposite responses to cropland losses and gains between western and eastern Europe. Effects of historical LCCs on European climate are non-negligible and region-specific, and ignoring land-climate biophysical interactions may lead to sub-optimal climate change mitigation and adaptation strategies.

[1] Industrial Ecology Programme, Department of Energy and Process Engineering, Norwegian University of Science and Technology (NTNU), Trondheim, Norway. [2] Department of Mathematical Sciences, Norwegian University of Science and Technology (NTNU), Trondheim, Norway. [3] National Tibetan Plateau Data Center, Institute of Tibetan Plateau Research, Chinese Academy of Sciences, Beijing, China. [4] State Key Laboratory of Earth Surface Processes and Resource Ecology, Faculty of Geographical Science, Beijing Normal University (BNU), Beijing, China. [5]These authors contributed equally: Bo Huang, Xiangping Hu. ✉email: francesco.cherubini@ntnu.no

Climate variability is a main driver of changes to landscapes and ecosystems[1,2], but also land cover changes (LCC) influence the regional climate[3,4] because they alter biophysical mechanisms like evapotranspiration, albedo, and surface roughness[5–7]. The temperature response to LCCs depends on the type of land cover[8–10] or management[11–13], background climate conditions[14–16], and spatial extension of the LCC[14,17]. Spatial and seasonal heterogeneities are also common[18,19]. For example, forest losses from 2003 to 2012 locally increased air temperature up to about 1 °C in temperate and tropical regions as a result of declined evapotranspiration[19], especially during summer[18,20]. On the contrary, cooling benefits of forests are smaller at high latitude, where the evapotranspiration benefits are compensated by lower surface albedo[14,19].

Satellite observations provide a valuable opportunity to study land cover properties and changes[21–23], and remotely sensed data are frequently used to generate maps of forest area extension and contraction[21,24], or development trends in cropland or grassland[25–28]. Because of the importance of land cover feedbacks to climate, an accurate description of the different types of vegetation cover and their changes over time is a key asset of climate models[22,29,30]. Many of the existing global land cover datasets available for modeling purposes have little consistency in terms of period of observation, spatial resolution, and accuracy[22,31]. In the recent years, the European Space Agency (ESA) Climate Change Initiative Land Cover (CCI-LC) produced a time-series (from 1992 to 2015) of high-resolution (300 m) land cover maps combining multiple remote sensing products and ground-truth observations[22,32]. These maps were specifically developed for reducing previous limitations and to advance a more realistic representation of land cover dynamics in climate models[22,31], and are also used to characterize temporal dynamics and spatial patterns of land cover changes at a landscape and global level[23,33].

Previous research mainly assessed the regional climate implications of individual land cover transitions, such as the effects of historical land use changes (mainly forest clearance)[18,20], changes in agricultural management[34–36], forest management[3,37], or idealized large-scale scenarios of deforestation/afforestation[4,38], but the combined effects from the mix of recent historical land cover changes in Europe have not been explored. Further, many previous assessments of impacts from LCCs focused on a single variable, typically temperature or precipitation[9]. However, bulb temperature alone is only a partial characterization of surface heat content, and the analysis of a joint temperature−humidity response offers a more complete measure of warming change[39]. The combination of temperature ($T$) and surface air moisture ($q$) into a single variable, called moist enthalpy or equivalent temperature ($T_E$), informs about near-surface atmospheric heating, a relevant measure for the human and ecosystem capacity to adapt to climate conditions[7,11,20,39,40].

Here, we use the CCI-LC dataset in combination with a regional climate model (the Weather Research and Forecasting model, WRF)[41] to quantify the effects that land cover differences have had on European climate between 1992 and 2015. Two simulations with the land cover in 1992 and 2015 (named LC1992 and LC2015, respectively) are performed and the resulting relative differences (LC2015–LC1992) in 2-m air temperature, surface air humidity, and equivalent temperature investigated. Another experiment (named NoCRP_AB) uses a modified land cover dataset where conversion of cropland to other land cover classes is intentionally omitted, but other land cover classes are allowed to translate to cropland according to the historical transitions. The difference between NoCRP_AB and LC2015 highlights the effects of cropland abandonment on regional climate. Unlike many previous studies that had to use one less realistic large-scale simulation for each LCC to single out its effects on regional climate, our analysis simultaneously considers the effects of the mix of historical land cover changes that recently occurred in Europe and then disentangles the individual contribution with a new approach based on a ridge statistical regression. This approach does not require the explicit consideration of the different components of the surface energy budget, and directly shows the temperature changes from each land transition. While acknowledging the limitations related to the use of a single regional climate model, model outputs are generally able to reproduce the spatial patterns of observational datasets and key results are typically consistent with other modeling and empirical estimates (see Methods).

## Results

**Land cover transitions.** From 1992 to 2015, around 70 Mha of land transitions occurred in Europe (Fig. 1; see Supplementary Fig. 1 for the maps of the main transitions). Approximately 25 Mha of agricultural land was left abandoned, which was only partially compensated by cropland expansion (about 20 Mha), resulting in about 5 Mha of net loss. Declines in agricultural land mostly occurred in favor of forests (15 Mha) and urban settlements (8 Mha). Institutional and socio-economic factors are found to be the most important drivers of agriculture abandonment in Europe, because it usually took place in areas without major constraints for crop production[27,42]. For example, the collapse of the former Soviet Union triggered extensive farmland abandonment as a result of declines in agricultural investments, exposure to global agricultural markets, and outmigration from rural areas[27]. The type of natural succession after agricultural abandonment depends on soil fertility, local climate, and nearby vegetation[43]. In the first years after abandonment (up to 5 years), a dense herbaceous cover usually develops on the land, followed by the growth of new vegetation to woodland and forests[42,44]. Other studies investigated the causes and effects of declines of cropland areas in Europe in more detail[25–27,42]. As a consequence of agriculture abandonment, forested areas in Europe increased by about 23 Mha, with about 7 Mha of net gain. Drying of wetland and peatland in northern Europe also contributed to forest area extension. Climate changes, such as higher evaporation from warmer summers, decreased precipitation, and increased runoff from melting snowpack, favor advances of the tree lines from the margins to the center of the wetland[45,46]. Another major land transition in Europe concerned urban sprawling, which mostly took place on agricultural land in proximity to densely populated areas as a result of population and economic growth[47].

**Annual mean changes in $T$, $T_E$, and $q$.** We found an annual average temperature change of $-0.12 \pm 0.20$ °C (mean ± standard deviation), with $-0.42$ and $+0.22$ °C as the 5th and 95th percentile (Fig. 2a), respectively, from recent LCCs in Europe. An average cooling is consistently spread in the southern, central and western part of Europe, whereas warming occurs in eastern Europe. Relatively little changes in temperature are observed in northern Europe (Scandinavia), where relatively limited land transitions took place. The continental average cooling is reflected by an average increase in latent heat ($+0.25 \pm 1.83$ W m$^{-2}$) and reduction in sensible heat fluxes ($-0.21 \pm 2.14$ W m$^{-2}$), although the spatial variability is high owing to the variety of land transitions involved (Supplementary Fig. 2).

Agriculture abandonment plays a central role for the climate effects of recent LCCs in Europe. When the transitions from cropland to other land classes are excluded, we found an annual average temperature change of $+0.10 \pm 0.19$ °C, with $-0.23$ and $+0.33$ °C as the 5th and 95th percentile, respectively (Fig. 2b). The cooling benefits are largely lost when agriculture

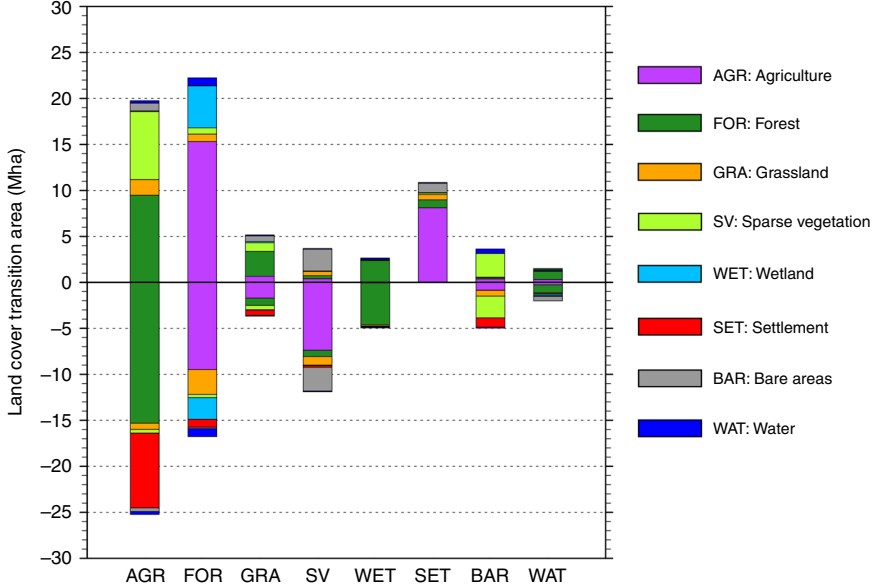

**Fig. 1 Land cover changes occurred in Europe between 1992 and 2015.** Positive values indicate an expansion of the respective land cover, negative values a contraction. The CCI-LC classification system is converted to the IPCC land classes to facilitate interpretation. See Supplementary Table 3 for the cross-walking table translating from one land classification system to the other. Supplementary Fig. 1 shows the maps of the major land cover transitions.

abandonment is masked (changes in latent heat fluxes decline to $-0.03 \pm 1.83$ W m$^{-2}$), and the net effects from the other land transitions are either warming or balancing out. An opposite response is observed in the eastern part of Europe, where exclusion of cropland abandonment shows a widespread temperature reduction (Fig. 2b). The same area was generally associated with warming in Fig. 2a. Contrary to other locations in Europe, in this subdomain, natural revegetation of agricultural land left abandoned is associated to higher local temperature. At an average European level, the analysis of the climate change signals through a probability function based on kernel density estimation shows the average cooling contributions of abandoned cropland (Fig. 2c). In the "LC2015–LC1992" case, the probability distribution of mean temperature changes peaks at around $-0.1$ °C (about 4.5% of the grids), whereas when conversion of cropland to other land classes is excluded ("NoCRP_AB–LC2015") the distribution is translated towards mean higher temperatures, and peaks at around $+0.1$ °C.

Changes in equivalent temperature $T_E$ are more pronounced than those in bulb temperature (Fig. 3). At a continental level, the average difference in $T_E$ from the recent LCCs is $-0.10 \pm 0.37$ °C, with $-0.58/+0.57$ °C as the 5th and 95th percentile, respectively. Both the mean value and spatial pattern are similar to that of $T$, but variability is larger. A contrasting response is still found between the western and eastern part of the domain, but for $T_E$ local mean annual differences can be up to $+1$ °C in eastern Europe and $-0.8$ °C in central Europe (Fig. 3a). Such an increase in equivalent temperature matches with the trends in surface humidity (Fig. 3b).

When agriculture abandonment is excluded (Fig. 3c), the average continental change in equivalent temperature is $+0.05 \pm 0.30$ °C ($-0.51$ and $+0.48$ °C as the 5th and 95th percentile, respectively). In this case, the reduction in surface air humidity drives the stronger cooling response of $T_E$ than $T$ in the eastern part of the domain. The analysis of the climate change signals through the probability density function clearly shows that $T_E$ has a larger distribution than $T$. For example, about 1% of the grids experience a cooling of $-0.7$ °C for $T_E$ (Fig. 3e), whereas it is $-0.5$ °C for $T$ (Fig. 2c). Further, while excluding agriculture abandonment tends to translate the climate change signal of $T$

towards a warmer climate, for $T_E$ it mainly affects the negative values of the distribution only, increasing the density of the grid cells that are slightly warmer or show no changes. This is driven by the decline in specific humidity from exclusion of agriculture abandonment (Fig. 3f).

In general, the results show a different climate system response between the western and eastern parts of Europe. This is linked to the importance that local conditions and background climate have in shaping how key components of the surface energy budget respond to LCCs[8,14,16]. In eastern Europe, the warming contribution from reduction in surface albedo after revegetation of abandoned cropland is stronger than the cooling benefits from larger latent heat fluxes associated with tree cover (Supplementary Fig. 2). This area has lower values of soil moisture than other places in Europe (Supplementary Fig. 3), thereby mitigating the potential for trees to dissipate evaporative cooling from the increased energy budget due to the decrease in surface albedo. The so-called soil moisture-temperature feedback refers to the additional warming occurring with shortages of evaporative cooling in regions affected by relatively dry conditions[48–50]. This is mainly observed in mid-latitude regions transitioning between wet and dry climates, where lower soil moisture availability directly impacts turbulent flux partitioning and surface temperature[49,51,52]. The eastern part of Europe is a transitional region between oceanic and continental climate, where the change from cropland to forests under soil moisture limitations leads to increases in surface temperature. This mechanism is confirmed by both modeling and observational analysis[49,53,54]. Other factors contribute to the observed spatial variability. Radiative properties of ecosystems are the major factors controlling surface temperature when incoming energy is the major limitation to vegetation growth or seasonal snowfalls are significant (such as in eastern Europe and boreal areas)[8]. Similarly, the tendency of forests to decrease surface temperature compared to open land is higher in relative proximity to the oceans, and gradually declines at increasing distances[14,19,55,56].

**Seasonality.** We found a seasonality in the intensity and spatial distribution of the regional temperature changes (Fig. 4). The averaged seasonal statistics at a continental level are smoothed by

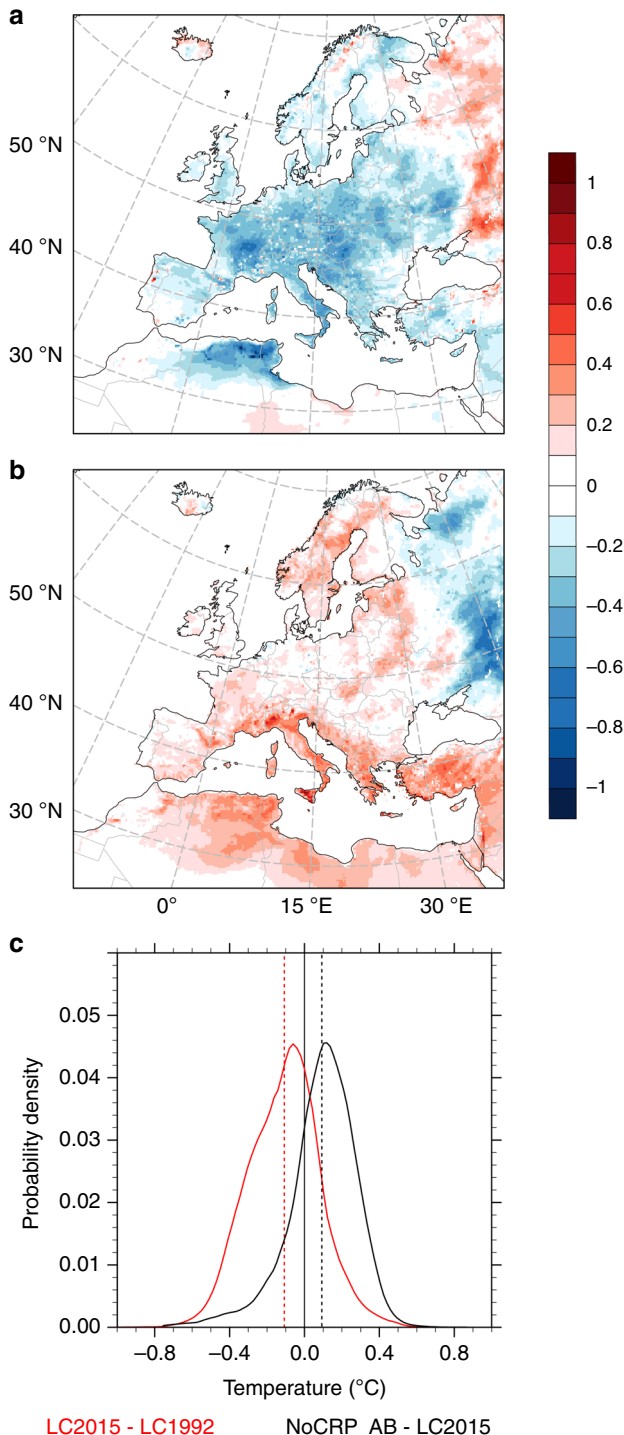

**Fig. 2 Annual average temperature changes from recent (1992–2015) land cover transitions in Europe.** Annual average temperature changes (°C) due to differences in land cover between 1992 and 2015 (**a**); annual average temperature changes excluding contributions from agriculture abandonment (**b**); probability density function of the annual average temperature changes across the entire domain (**c**). In **c**, the solid line shows the distribution of the values and the dashed vertical line indicates the respective mean value. LC1992 and LC2015 refer to regional climate simulations with the European land cover in 1992 or 2015, respectively. NoCRP_AB refers to a land cover dataset where agricultural land in 1992 is not allowed to convert to other land classes.

the contrasting effects of the mixed LCCs (Supplementary Fig. 4), but spatial variations can be appreciated. The seasonality is mainly the result of the interlinked combination of the annual variability in solar radiation, local conditions (e.g., snow cover, soil moisture), surface albedo and evaporation efficiencies among the land cover classes[18,19]. In winter, LCCs caused lower differences in temperature, except for the central- and north-eastern part of the domain where changes are more evident, but a clear spatial pattern does not emerge (Fig. 4a). Larger differences are found in spring (Fig. 4b) and summer (Fig. 4c), where a cooling of more than −1.0 °C is observed in some areas of central Europe. Colder temperatures are generally found in western and southern Europe as well. At the same time, spring and summer temperatures increased in eastern Europe. The mean annual warming previously observed in the eastern part of the domain is thus primarily driven by warmer spring and summer temperatures. This area is affected by seasonal snow cover in winter and early spring, and LCCs generally caused reductions in surface albedo from snow-masking effects in the central-east (from vegetation of abandoned cropland) and in the north-east (from vegetation of wetland). This has increased the amount of energy at the surface, especially during spring when solar radiation is larger. In summer, the seasonal warming in the central-east is weaker than in spring, because changes in surface albedo are smaller and evapotranspiration efficiencies of land cover classes play a major role. Higher summer temperatures in the north-eastern part are mainly due to wetland drying, where the energy dissipated as sensible heat instead of latent heat increases (wetland typically have the largest latent heat fluxes[57]).

A seasonality is also observed in the different extension of the spatial autocorrelation of the climate response represented by the spatial correlogram (the autocorrelation index plotted against distance). The correlation among pairs of spatial observations is positive up to about 150 km, but there is a strong seasonality in the decrease of the autocorrelation index at increasing distance between the data (Supplementary Fig. 5). This decrease occurs at higher rates in winter than summer. In winter, the autocorrelation vanishes at less than about 50 km, while in summer it remains positive up to about 200 km. This aligns with the more consistent pattern of the European climate response in summer than winter (Fig. 4).

The monthly mean climatological differences in $T$, $T_E$ and $q$ at the 2-m level induced by the recent LCCs in Europe are shown in Fig. 5. In terms of monthly mean values of the individual variables, $T$ and $T_E$ generally exhibit similar seasonal patterns, but $T_E$ values are larger (see Supplementary Fig. 6). During winter and early spring, humidity is low and differences between the two variables are small. As humidity increases from late spring to early fall, differences become larger. The comparison of the monthly mean differences between LC1992 and LC2015 across the whole domain shows that $T$ generally decreases up to a maximum in late spring and summer, but trends in specific humidity and $T_E$ are less clear due to the large spatial variability that tends to balance out locations with warming or cooling (Fig. 5a). For example, the average difference in July of $T_E$ is +0.002 °C, but with a standard deviation of about 1 °C. The maximum difference between the changes in $T$ and $T_E$ from the recent LCCs occurs in summer.

The spatial distribution of the changes in equivalent temperature shows a stronger seasonal response than $T$ (see Fig. 5b, d for winter and summer, and Supplementary Fig. 7 for spring and autumn). In particular, higher summer warming (>1 °C) is found in the eastern part of the domain (Fig. 5d), where increases in surface humidity are also at a maximum (Fig. 5e). This result can be explained by increased vegetation greenness and leaf area index (especially during the growing season), which is associated

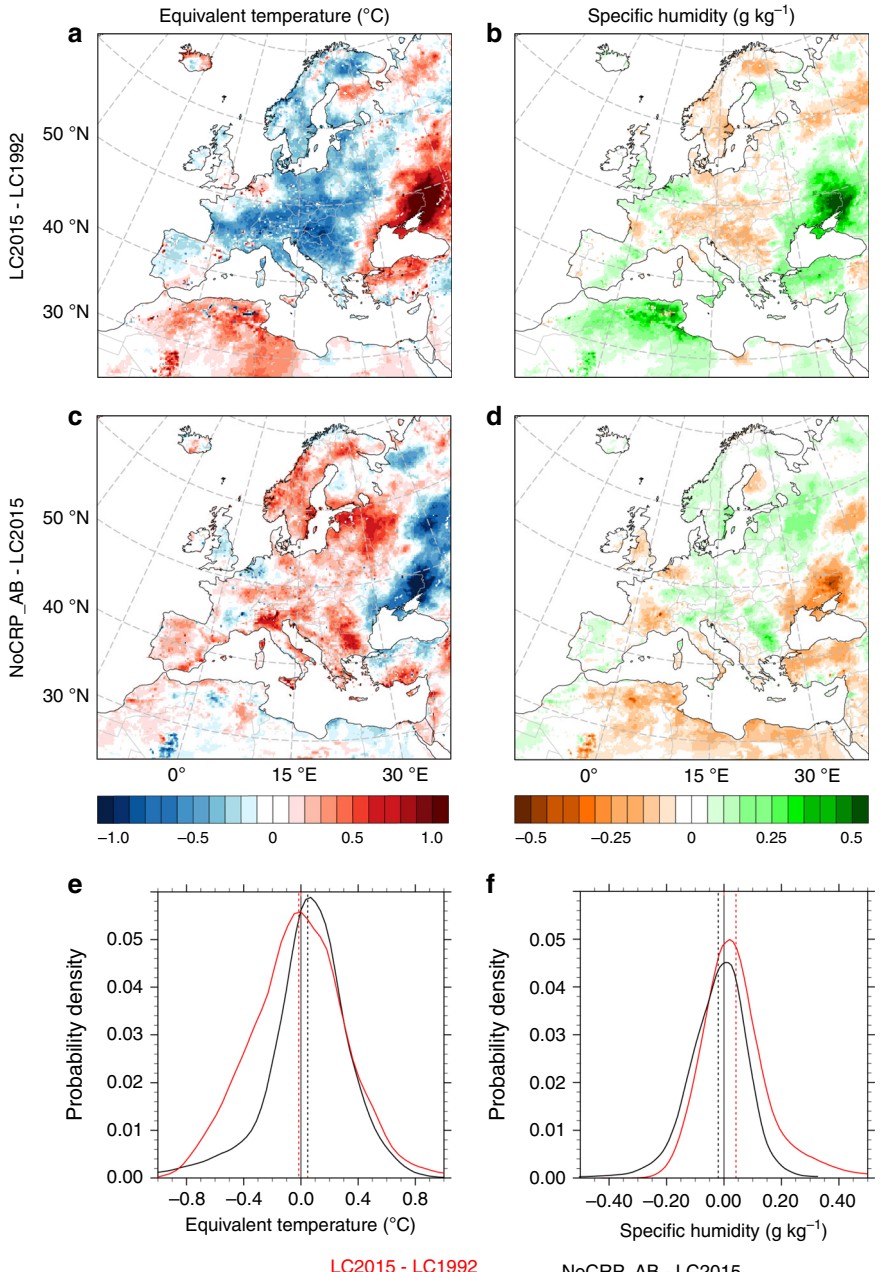

**Fig. 3 Annual average changes in equivalent temperature ($T_E$) and absolute specific surface humidity ($q$) from recent (1992–2015) land cover transitions in Europe.** Annual average $T_E$ changes (in ºC) due to the mix of land cover transitions (**a**) and from excluding contributions of cropland abandonment (**c**); probability distribution function of $T_E$ changes across the entire domain (**e**); annual average $q$ changes (in g kg$^{-1}$) due to the mix of land cover transitions (**b**) and from excluding contributions of cropland abandonment (**d**); probability distribution function of $q$ changes across the entire domain (**f**). In **e** and **f**, the solid line shows the distribution of the values and the dashed vertical line indicates the respective mean value. LC1992 and LC2015 refer to regional climate simulations with the European land cover in 1992 and 2015, respectively. NoCRP_AB refers to a land cover dataset where cropland in 1992 is not allowed to convert to other land classes.

with higher physical evaporation and transpiration rates. The major transitions in this region are cropland to forests, and in the northern part drying of wetland. $T_E$ correlates with vegetation activity stronger than $T$, especially during the growing season, and both moisture availability and equivalent temperature are usually larger in forested areas[40].

**Decomposition to individual land transitions.** A ridge-regression approach was used to disentangle mixed temperature signals and identify individual contributions for each LCC

(Fig. 6). This procedure allows direct quantification of the temperature effects correlated with each land transition from simulations where multiple LCCs are modeled simultaneously, i.e. without the need to (i) individually run one (usually unrealistic) area-extended simulation per LCC, or (ii) indirectly infer temperature changes by post-processing and re-assembling the components of the surface energy budget. In this approach, the abundance of each specific land transition determines the robustness of the estimate, and values that are not statistically robust are filtered out (see Methods). At an average European level, transitions from forest to any other land cover class show

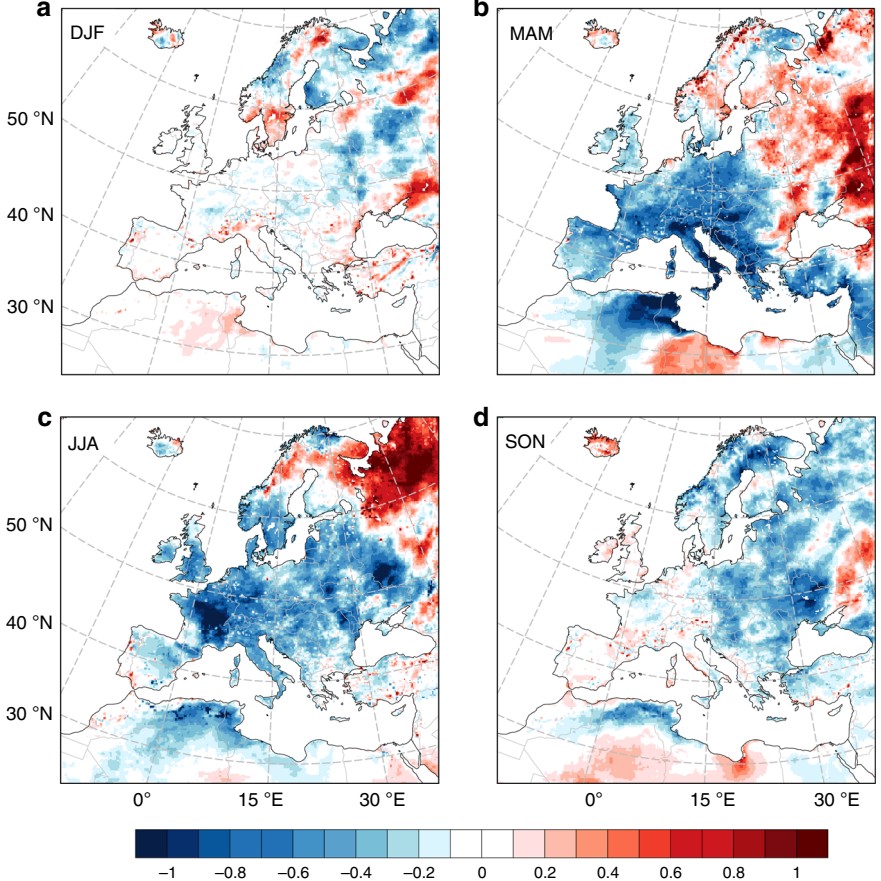

**Fig. 4 Seasonality of the temperature changes from recent land cover changes in Europe.** Average temperature changes (°C) in winter (**a**), spring (**b**), summer (**c**) and autumn (**d**). December, January and February (DJF); March, April May (MAM); June, July and August (JJA); September, October and November (SON).

mean warming effects, including conversion of forests to cropland ($+0.15 \pm 0.03$ °C), grassland ($+0.23 \pm 0.06$ °C), and urban ($+0.27 \pm 0.06$ °C). Urban sprawling always shows warming contributions, irrespective of the type of previous land cover. Although a direct comparison is difficult owing to the uncorrelated behavior of $T$ and $q$, changes in $T_E$ are usually larger than those in $T$, especially for transitions involving evergreen needle leaf forests that have low evapotranspiration rates. $T$ and $T_E$ have different signs mainly for transitions including open shrubland or grassland. These land cover classes have the smallest average difference between $T$ and $T_E$, which on the other hand are the largest for cropland, wetland and forest[40]. The transitions between the former and the latter groups of land cover classes thus yield the major marginal differences between $T$ and $T_E$. For example, transitions from wetland to shrubland typically show reductions of $T_E$, despite increases of $T$. This occurs because wetland usually converts more solar energy into latent heat than sensible heat ($q$ is thus relatively high)[58]. The reduction in $q$ after the wetland-to-shrubland transition can overwhelm the corresponding increase of $T$, and the transition results in negative $T_E$ values. This LCC is usually associated with an increase in surface albedo values as well, which reduces the amount of solar energy to be dissipated at the surface through evapotranspiration.

Potential gradients in the regional climate effects from LCCs are explored across the two major subdomains observed above, i.e. central and western Europe (subdomain A) and eastern Europe (subdomain B). The main differences mostly concerned the contrasting response to transitions involving forest cover. In subdomain A, conversion of evergreen or deciduous forest to

cropland results in an average warming of $+0.21 \pm 0.03$ °C ($T_E = +0.20 \pm 0.05$ °C) and $+0.12 \pm 0.03$ °C ($T_E = +0.28 \pm 0.09$ °C), respectively. In subdomain B, these transitions are associated with an average cooling of $-0.14 \pm 0.05$ °C ($T_E = -0.15 \pm 0.07$ °C) and $-0.10 \pm 0.05$ °C ($T_E = -0.02 \pm 0.09$ °C). This is mostly due to the local conditions discussed above, such as the interplay between surface albedo changes, evapotranspiration efficiencies and soil moisture.

We evaluated our estimates of temperature changes from specific LCCs against recently available land surface temperature observations. A direct comparison is performed with a recent empirical dataset derived from potential vegetation changes where a space-for-time approximation is applied to multiscale remote sensing products for the period 2008–2012[8]. This dataset summarizes estimates of the main components of the energy budget (over areas affected by land cover changes only), including day and night changes in land surface temperature, for a variety of LCCs at 1° resolution[8,59]. We found that our estimates are broadly consistent with the observational dataset (Supplementary Table 1). Considering the land cover transitions available in both studies, 52% of them are within the respective uncertainty ranges, and most of the remaining are outside their uncertainty intervals but still have the same sign. Results are generally more consistent for transitions between forests and cropland, whereas variability is larger when grassland is involved. In line with our study, the observational dataset also finds an average warmer surface in presence of trees than open land in subdomain B, which is the opposite of the response found in subdomain A. The contrasting response between eastern and western Europe is reported by

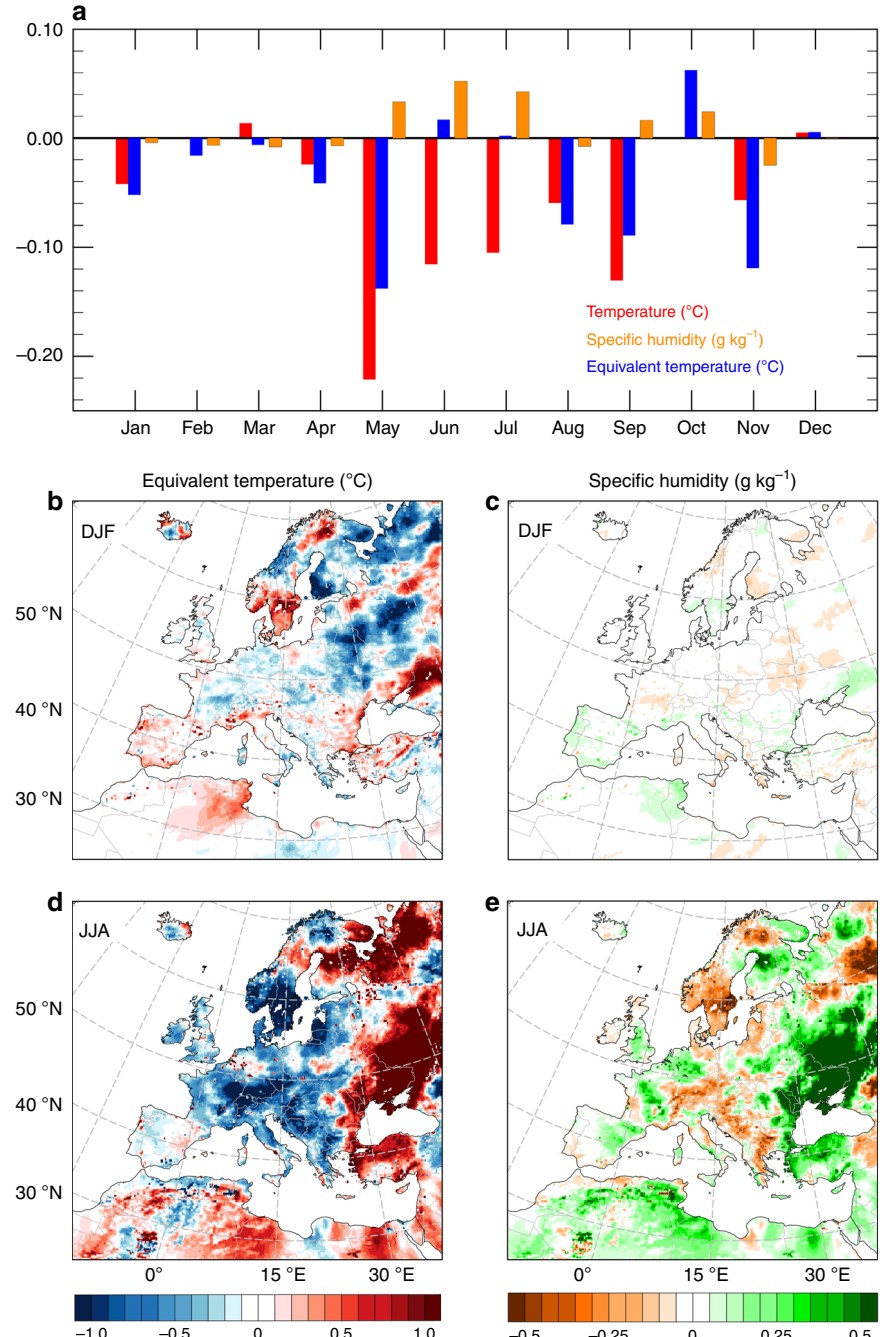

**Fig. 5 Seasonality of the changes in equivalent temperature and specific air humidity from the recent land cover changes in Europe.** Comparison of the mean monthly differences (LC2015–LC1992) in air temperature, equivalent temperature, and specific air humidity (**a**); average equivalent temperature changes (°C) in winter (**b**) and summer (**d**); average specific humidity changes (g kg$^{-1}$) in winter (**c**) and summer (**e**). December, January and February (DJF); June, July and August (JJA).

other studies based on pair-site analysis, which used remotely sensed daily average surface temperature differences between adjacent forestland and open land[19,56].

More generally, our results can also be compared to previous studies that modeled temperature changes for a specific LCC (with forestland-open land as the most common transitions)[9,19,56]. Compared to the numerical values summarized in a recent review[9], we find that our figures are typically at the lower end of the reported ranges, and are closer to observations than other modeling studies. This is likely due to the smaller spatial extension of the land transitions in our analysis compared to those simulated

in other modeling studies, which are usually based on extensive and homogeneous changes in land cover. Our results are the outcome of historical small and mixed LCCs and represent the direct regional effects. The response is expected to be stronger for large-scale transitions that can trigger more substantial direct and indirect effects[9,14,60]. A similar validation of our estimates for changes in $T_E$ was not possible owing to a lack of corresponding observational datasets. Future modeling and empirical studies can try to increase our understanding of the underlining drivers that regulate changes in $T_E$ with land cover types and increase the robustness of specific estimates.

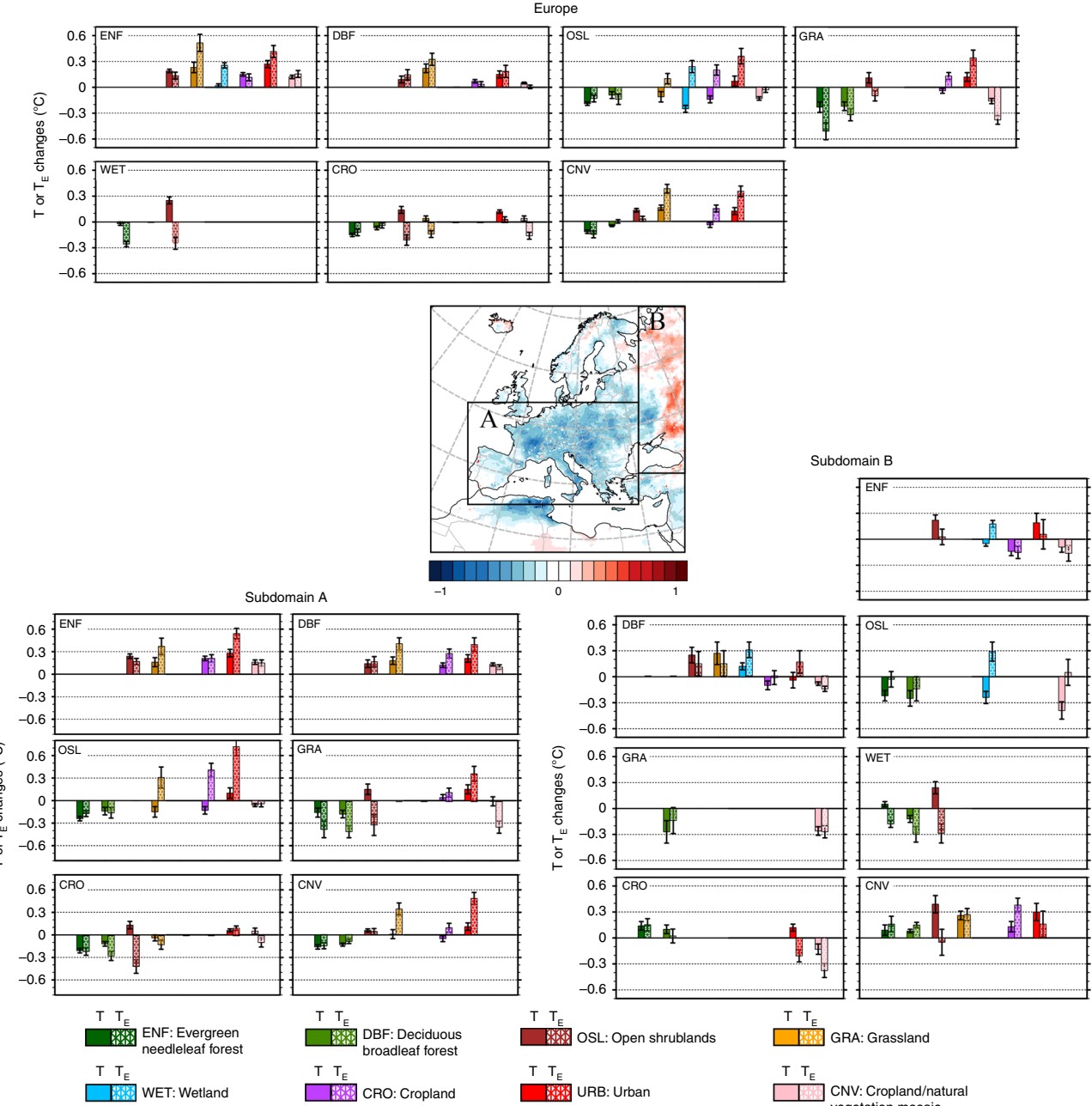

**Fig. 6 Decomposition of the temperature changes to individual land cover changes.** Temperature and equivalent temperature change signals are shown for the European domain and its major subdomains. In each subpanel, the transitions are to be read from the land class indicated in each top left corner to the different land cover classes indicated by the different colors in the horizontal axis. The full bar indicates changes in $T$, the bar filled with texture $T_E$. The whisker indicates the standard error around the mean value. The map in the center shows the locations of the two subdomains and the color scale reproduces the annual mean temperature response (in °C) as in Fig. 2a.

## Discussion

This study uses a state-of-the-art regional climate model to show how the mix of recent historical land cover changes in Europe can influence the regional climate. The distribution and variability of land cover types influence temperature and moisture availability in the lower atmosphere. Revegetation after agriculture abandonment is the main process found to be associated with biophysical cooling at a continental scale, but it has an opposite effect in the eastern part of the domain. This is a climatic transition zone where tree cover expansion shows a biophysical warming mostly due to the soil moisture–temperature feedback and other local factors like reductions in surface albedo. Whereas most previous studies had to rely on simulations of less realistic and large-scale changes of one land cover class to single out the corresponding climate effects, our results are produced from the simultaneous modeling of a variety of historical small-scale and mixed changes in land cover between 1992 and 2015. Specific climate impacts are directly correlated to individual land transitions with the ridge-regression approach that is used for the unmixing of the temperature effects. The approach does not require thresholds or constraints, because it decomposes the effects of the various LCCs by using the actual land cover data within each grid cell as predictors, and can be adapted to different model outcomes, variables and simulation domains.

Changes in vegetation cover are modeled using the recent ESA CCI land cover dataset that is specifically designed for climate modeling studies. CCI-LC maps are found to improve characterization of the recent global and regional land cover changes relative to other datasets[31]. In general, our regional climate simulations with either LC1992 or LC2015 reduce both model bias and the root-mean-square error when compared to simulations with the default land cover in WRF (see Methods and Supplementary Fig. 8). Due to the settings of our experiments (i.e., simultaneous modeling of multiple LCCs per grid cell), the changes in the components of the surface energy budget for each individual LCC cannot be directly compared with those available from observation-driven studies. However, the comparison was possible for the final temperature responses associated with each LCC, and results were broadly consistent with observational datasets. It was also possible to validate key variables such as soil moisture and surface albedo, which were in line with observational data for both average numerical values and spatial patterns (Supplementary Fig. 3 and Table 2). Direct comparison of results between modeling and empirical studies should always consider the potential differences in methodological approaches among the available datasets, such as the land cover classification system, the spatial resolution, and the temporal scale of analysis. For example, remotely sensed products measure land surface temperature, while modeling studies focus on 2-m air temperature (see Methods). Air temperature is usually dependent on land surface temperature[61], but satellite retrievals of land surface temperature only occur under clear-sky conditions, and correlation between the two temperature indicators varies in time and across different land covers[62]. Further, the use of a single model can have inherent limitations, and future model inter-comparison efforts are required to better assess model uncertainties and constraint dependencies of results on individual model configurations[15,63]. Future research can improve the representation of plant physiology and vegetation dynamics, such as forest development stages, within both remotely sensed datasets of land cover classes and climate models[29,59]. There are differences in vegetation structure within the same land cover class, and the possibility to explicitly map and model gradients in forest structures can lead to more accurate modeling of surface fluxes in forests. For example, inferring the fractional cover of the major plant functional types and vegetation properties within a land cover class directly from satellite radiances, and coupling their representation to regional climate models, is an option for more confident modeling of vegetation cover.

Our estimates of temperature effects from LCCs were produced under present climate conditions, but future impacts of land cover changes on regional climate can vary under a different future background climate[16]. Vegetation phenology can respond to climate feedbacks such as higher $CO_2$ concentration, increases in average temperature and frequency of extreme events, and changes in precipitation patterns[64,65]. For example, the southern part of subdomain B and other places in Southern Europe will likely experience a drier climate in the next decades in the absence of ramping global climate change mitigation efforts[66]. This could further reduce soil moisture, and through the soil moisture–temperature feedback change the potential response of a certain area to tree cover expansion. Overall, in addition to the inherent uncertainty of future regional projections of climate change[67], the vegetation response to climate feedbacks is highly complex and difficult to quantify, and parallel developments of modeling and empirical approaches are needed to refine land management strategies by anticipating variability in future regional climate.

The regional or local scale is the scale at which most decisions about land management are taken or implemented, and the

availability of site-specific metrics on land-climate interactions is central to deploy effective land-based climate change mitigation and adaptation strategies[9,68,69]. This study quantified the regional temperature effects of recent land cover changes in Europe with a regional climate model, and, by directly connecting local climate impacts to individual LCCs, it expands the scientific basis towards a better understanding of the potential land-climate interactions for more climate-oriented land management strategies. Land-climate interactions are strictly coupled to societal developments, especially in terms of urbanization and cropland extension/contraction. Reduction of urban sprawling will bring temperature benefits across all Europe. On the other hand, the climate benefits of revegetation of abandoned cropland is conditional to locations, because agriculture abandonment cools local climate in many places of western and continental Europe, but it has an opposite effect in eastern Europe. The local conditions for land processes related to regional climate, vegetation cover, and water availability needs to be considered when assessing LCCs. This calls for region-specific integrated approaches of land management strategies to embrace both scientific and socio-economic contexts.

Current international climate policy frameworks are solely based on greenhouse gases[69], and do not include biophysical effects on regional climate. Interactions between land cover and the regional and local climate system should be more prominently considered in land management planning, because they offer the potential to codeliver regional-scale climate adaptation and mitigation objectives. The availability and progressive consolidation of simplified metrics for land-climate responses, and further development of remote sensing products coupled to regional climate models to refine estimates, will be instrumental to cover this gap. LCCs also influence ecosystem services other than climate regulation, such as their productivity, water holding capacity, and biodiversity[70]. Sustainable land management policies from the local to the regional level should prioritize the development of consistent approaches to embed these multiple dimensions of land use.

## Methods

**Land cover dataset**. The European Space Agency (ESA) Climate Change Initiative (CCI) land cover (LC) product is used to map changes in vegetation cover. The ESA CCI-LC maps are provided at a spatial resolution of 300 m for a period of 24 years, from 1992 to 2015[32]. These maps characterize the global land surface using 37 land cover classes based on the United Nations Land Cover Classification System (UNLCCS)[71], and were designed to overcome previous limitations and reduce uncertainty in the representation of land cover and LCCs in climate models[22,31,32]. The dataset was produced after combination of the global daily surface reflectance of five different satellite observation systems, with the ambition to keep high levels of consistency over time. The accuracy of the CCI-LC products was assessed at a global level using an independent validation dataset, and estimated to be of 75.4%[32]. The highest accuracy was found for cropland classes, forests, urban areas, bare areas, water bodies and perennial snow and ice. Mosaic classes, lichens and mosses showed the lowest accuracy.

We used three types of land cover data to explore the effects of recent land cover changes in European climate and single out the role of cropland abandonment: LC1992, LC2015, and NoCRP_AB. LC1992 and LC2015 represent the land cover in Europe in 1992 and 2015, respectively. NoCRP_AB is a land cover dataset where the IGBP classes cropland and cropland/natural vegetation mosaic in 1992 are not allowed to convert to other land classes. However, other land classes are allowed to convert to cropland and cropland mosaic according to the historical transitions. This means that, between 1992 and 2015, cropland and cropland/natural vegetation mosaic are only allowed to expand, and not to contract.

In order to facilitate the interpretation of the land transitions that occurred in Europe (Fig. 1), the 37 UNLCCS classes were aggregated into the generic IPCC land classes, according to the specific cross-walking table provided by the ESA CCI-LC products (reproduced in Supplementary Table 3)[32].

**Regional climate simulations**. Regional climate simulations are performed with the Weather Research and Forecasting (WRF) model version 3.9.1[41]. WRF is a next-generation mesoscale model that is suitable for operational research across scales. WRF produces simulations based on actual atmospheric conditions (i.e.,

from observations and analyses), and it has been validated to capture the spatio-temporal patterns of climate against observations in Europe[72–74].

The specific configuration of WRF used in this study follows the settings of the international Coordinated Regional Climate Downscaling Experiment (CORDEX) initiative (EURO-CORDEX)[74]. The initial and lateral boundary conditions are from the European Centre for Medium-Range Weather Forecasts Interim reanalysis (ERA-Interim)[75]. The physical parameterization schemes include the Thompson microphysics scheme[76], the Rapid Radiative Transfer Model for GCMs longwave and shortwave radiation[77], the Mellor-Yamada Nakanishi and Niino boundary layer scheme[78], the Kain-Fritsch convection parameterization[79], and the Community Land Model version 4.0 (CLM4) with IGBP-MODIS land use classification[80,81].

CLM4 is a state-of-the-science land surface process model that represents several aspects of the land surface, including surface heterogeneity, and consists of components related to land biophysics, hydrologic cycle, biogeochemistry, human dimensions, and ecosystem dynamics. CLM4 in WRF has a detailed description of land surface, in which the vertical structure includes a single-layer vegetation canopy, a five-layer snowpack, and a ten-layer soil column[41]. In each grid cell, the land surface is categorized into five primary subgrid land-units (glacier, lake, wetland, urban, and vegetated) that share the same atmospheric forcing and flux feedback to the atmosphere within a grid cell. The surface variables are calculated by averaging the subgrid quantities weighted by their fractional areas (tile approach). The urban land-unit uses the "urban canyon" concept[82] to represent the canyon geometry, described by building height and street width. The vegetated subgrid is comprised of up to 15 plant functional types (PFTs) that differ in structure and physiology as leaf and stem optical properties, root distribution parameters, aerodynamic parameters, and photosynthetic parameters[41]. These parameters are monthly prescribed and daily updated by linearly interpolating monthly values[83]. CLM4 includes new treatments of soil column-groundwater interactions, soil evaporation, aerodynamic parameters for sparse/dense canopies, vertical burial of vegetation by snow, snow cover fraction and aging, black carbon and dust deposition, and vertical distribution of solar energy. The CLM two-stream radiation model calculates the model equivalent surface albedo using climatological monthly soil moisture along with the vegetation parameters of PFT fraction, leaf and steam area index. Several PFTs can coexist in a given grid cell, and the energy balance and surface fluxes are calculated at the PFT level before being aggregated at the grid-scale level based on the proportion of PFTs in the grid cell. As WRF reads as input IGBP-MODIS/USGS LC classifications, we use a cross-walking table[8] (Supplementary Table 4) to translate the 37 classes in CCI LC to the 21 classes of the IGBP-MODIS classification system, which CLM4 then translates to PFTs. The distribution to PFTs is a well-known potential source of uncertainty, especially for mixed classes and northern boreal forests (an area where little LCCs occurred in our study)[29]. The use of cross-walking tables is a common viable approach to ensure transparency and reproducibility of model results until the mapping of plant functional traits at global scale will become possible and made available[22,29].

Due to limitations in terms of computational time, a 24-year (from 1 January 1992 to 31 December 2015) averaged dataset is produced from the ERA-Interim data and used as initial and lateral boundary conditions. This new dataset contains 6-hourly data from 1 January at 0000 UTC and run through 31 December at 1800 UTC for a period of 1 year. The land cover datasets are aggregated at a horizontal resolution of 0.11° (ca. 12 km, the highest resolution available from EURO-CORDEX) retaining the proportions of the different land classes per grid cell. Three independent WRF simulations are then performed with the three land cover datasets LC1992, LC2015, and NoCRP_AB. Since lateral boundary conditions do not vary between experiments, the resulting differences in model outcomes can be attributed to the different land cover datasets. The simulations are performed with 40 atmospheric levels and an integration time step of 72 s. The first 15 days are treated as model spin-up time and therefore excluded from the analysis. The analysis also excludes 30 simulation grids (ca. 360 km) from the border of the EURO-CORDEX domain to clear the noise in the lateral boundary conditions.

The simultaneous contribution of 2-m air temperature ($T$) and absolute specific humidity ($q$) is computed in terms of equivalent temperature ($T_E$), which indicates the temperature a sample of air would have if all its latent heat was isobarically converted to sensible heat. It can be estimated through the following equation[39,40]: $T_E = T + Lq/C_p$, where $L$ is the latent heat of vaporization and $C_p$ the specific heat of dry air.

**Spatial correlogram**. Spatial correlograms are used to study the spatial auto-correlation of the changes in temperature. This is an approach used to show how correlated are pairs of spatially distributed observations at increasing distance between them[84,85]. The spatial autocorrelation coefficient is computed for each distance class, and it is usually measured by the Moran's $I$ statistics as follows[86–88]:

$$I(d) = \frac{\frac{1}{W} \sum_i^n \sum_j^n w_{ij}(y_i - \bar{y})(y_j - \bar{y})}{\frac{1}{n} \sum_{i=1}^n (y_i - \bar{y})} \quad \text{for } i \neq j, \tag{1}$$

where $y_i$ and $y_j$ are the values of the changes in temperature in grids $i$ and $j$. A matrix of geographic distances needs to be calculated for all pairs of locations. We then convert these distances to classes $d$. The weight factor $w_{ij}$ is 1 when the pairs of sites belong to distance class $d$, and 0 otherwise. $W$ is the number of pairs in the

computation for a given distance class, and it is equal to the sum of $w_{ij}$ for that class. Moran's $I$ takes values in the interval $[-1, 1]$ and can be interpreted as the Pearson's correlation coefficient. Positive values of $I$ indicate positive auto-correlation and negative values of $I$ indicate negative autocorrelation[85].

**Temperature changes of individual land transitions**. A ridge-regression approach is introduced to retrieve the effects of the individual land cover transitions on the temperature and equivalent temperature changes. Although the methodology and applications differ, the concept is similar to the one recently used to characterize the changes in surface properties and energy fluxes from specific vegetation cover changes[8,59].

To identify the signal of the temperature changes due to the individual land cover transitions, we use a set of nonoverlapping local windows that covers the domain to unravel the effect on temperature from each LCC. Only the local effects of land cover transitions inside the window are considered, and the indirect perturbations due to regional changes outside the window are ignored. Local effects dominate the overall biophysical impacts for space-limited land cover transitions[89], such as those recently occurred in Europe.

The size of the local window is 5-by-5 grids, approximately 60-by-60 km, in which the local climate is assumed to be uniform. Climatic gradients that result from topographical differences are masked according to refs. [8,59]. To unmix the temperature signals resulting from the mix of the possible land cover transitions among the various classes, a linear regression approach is applied separately to the $N$ windows,

$$y_i = X_i \beta_i + \varepsilon_i, \, i = 1, 2, ..., N, \tag{2}$$

where for window $i$, $X_i$ is the explanatory variable, i.e. a matrix containing the fractions of the transitions of all the land cover classes for the 25 grids of each window (with the first column made of ones to capture the intercept), $y_i$ is a vector that contains the 25 values of changes in temperature, $\varepsilon_i$ is the error term referring to the model residuals, and $\beta_i$ is the vector of the regression coefficients. The residuals, $\varepsilon_i$, $i = 1, 2, ..., N$, are assumed to be independently and identically distributed with mean 0 and variance $\sigma_{\varepsilon,i}^2$. Equation (2) can be explicitly written for window $i$ as a system of equations:

$$\begin{aligned} y_1 &= \beta_0 + \beta_1 x_{11} + \beta_2 x_{12} + \cdots + \beta_m x_{1m} + \varepsilon_1 \\ y_2 &= \beta_0 + \beta_1 x_{21} + \beta_2 x_{22} + \cdots + \beta_m x_{2m} + \varepsilon_2 \\ &\vdots \\ y_n &= \beta_0 + \beta_1 x_{n1} + \beta_2 x_{2n} + \cdots + \beta_m x_{nm} + \varepsilon_n, \end{aligned} \tag{3}$$

where the dependence on $i$ is dropped to simplify the notation. $x_{nm}$ is the fraction of change in land class $m$ in grid $n$, $n$ is the number of grids of the local window ($n = 25$) and $m$ is the number of land cover classes ($m = 14$). The aim is to find the difference in regression coefficients between each pair of land classes to inform about the effect of the land transition on temperature by solving the system of Eq. (3). Once the system is solved, we can use the coefficients $\beta_i$ for window $i$ to understand the local temperature effects of a given transition of land cover classes from $j$ to $k$ by setting $x_j = -1$ and $x_k = 1$, and all the other $x$ to 0.

However, not all coefficients in $\beta_i$ can be estimated simultaneously since the transitions $x_{jk}$, $k = 1, 2, ..., 14$, in each grid in window $i$ sums to zero. In addition, in some cases, there are zero-columns in the matrix of the explanatory variable when transitions between specific land classes are missing, and the corresponding regression coefficient cannot be estimated. This is handled by solving Eq. (3) for each window $i$ using ridge regression to stabilize the estimation. This is a robust method frequently applied in statistical regression[90–92]. The solution for window $i$ is given by

$$\hat{\beta}_i = \left(X_i^T X_i + \lambda I\right)^{-1} X_i^T y_i, \tag{4}$$

where $\lambda$ stabilizes the inference when there is little or no information about a coefficient. We choose $\lambda = 10^{-12}$ so that a large variance is assigned to a coefficient for which the data do not inform. We investigate the expected change in temperature associated with each type of land cover transition as follows:

$$\Delta y_{i,j \to k} = \hat{\beta}_{i,k} - \hat{\beta}_{i,j}, \quad j,k = 1, 2, ..., m. \tag{5}$$

For a window $i$ where the data inform about the transition from land cover $j$ to land cover $k$, $\Delta y_{i,j \to k}$, takes a meaningful value, whereas when the data do not inform about the transition, the ridge regression leads to a zero value.

To distinguish between windows where the data inform about the transition of interest and windows where the data have little or no information about the transitions, we calculate the associated uncertainty of $\Delta y_{i,j \to k}$. We first calculate the estimate of residual variance

$$\hat{\sigma}_{\varepsilon,i}^2 = \left(y_i - X_i \hat{\beta}_i\right)^T \left(y_i - X_i \hat{\beta}_i\right)/\text{df}_i, \tag{6}$$

where the degrees of freedom (df$_i$) for window $i$ is $n$ minus the number of independent linear combinations of $\beta_i$ that the data inform about in that window. "T" is the transpose of a matrix. Using the estimated residual variance, we can

calculate the estimated covariances between coefficients as

$$\hat{\Sigma}_i = \left(X_i^T X_i + \lambda I\right)^{-1} \hat{\sigma}_{\varepsilon,i}^2. \tag{7}$$

The variances of the coefficients in $\hat{\beta}_i$ are presented in the diagonal of the covariance matrix $\hat{\Sigma}_i$, and the cross-covariance between the coefficients are stored in the off-diagonal parts of the covariance matrix $\hat{\Sigma}_i$. This provides the uncertainty associated with a temperature change due to a given land cover transition, $\Delta y_{i,j\rightarrow k}$, by calculating the variance:

$$\hat{\sigma}_{i,j\rightarrow k}^2 = \hat{\Sigma}_{i,jj} + \hat{\Sigma}_{i,kk} - 2\hat{\Sigma}_{i,jk}, \tag{8}$$

where $\hat{\Sigma}_{i,jk}$ denotes row $j$ and column $k$ of $\hat{\Sigma}_i$, i.e. $\hat{\Sigma}_{i,jj}$ and $\hat{\Sigma}_{i,kk}$ stand for the estimated variances of the estimated regression coefficients $\hat{\beta}_{i,j}$ and $\hat{\beta}_{i,k}$, and $\hat{\Sigma}_{i,jk}$ is the cross-covariance between $\hat{\beta}_{i,j}$ and $\hat{\beta}_{i,k}$. The covariance term is needed because the estimates of the land cover effects are not independent.

As described above, the system of Eq. (3) is solved in each of the local windows that spans all the domain. Within the window, water bodies are masked out and only the land cover transitions are considered. In addition, we also filter out windows with no land cover transitions. The regressions are then applied to all the IGBP land cover classes. This method leads to antisymmetry or skew-symmetry in the result, i.e., $\Delta y_{j\rightarrow k} = -\Delta y_{k\rightarrow j}$.

We then pool the information across the windows contained in a region (either Europe or two subdomains) to achieve informative estimates with useful uncertainty, because only a few of the $14 \times 13$ possible land cover transitions occurred in each window. The mean values for each land cover transition for all the windows which are considered in the regressions are calculated using the weight mean since it has the lowest variance

$$\Delta y_{j\rightarrow k} = \sum_{i=1}^{N} \frac{\Delta y_{i,j\rightarrow k}}{\hat{\sigma}_{i,j\rightarrow k}^2} \bigg/ \sum_{i=1}^{N} \frac{1}{\hat{\sigma}_{i,j\rightarrow k}^2} \tag{9}$$

and the variance is

$$\hat{\sigma}_{j\rightarrow k}^2 = 1 \bigg/ \sum_{i=1}^{N} \frac{1}{\hat{\sigma}_{i,j\rightarrow k}^2}, \tag{10}$$

where $N$ stands for the number of windows for regression. The windows where the estimated variance $\hat{\sigma}_i^2$ is unreliable are omitted, such as the variance is bigger than 100.

The mean and the median of the empirical residuals, $\hat{\varepsilon}_i = y_i - X_i\hat{\beta}_i, i = 1, 2, \ldots, N$, are almost zero, and the histogram of the empirical residuals is unimodal, symmetric, and similar to a normal distribution. This suggests that a Gaussian distribution can be assumed for the residuals so that confidence intervals for the land cover transitions can be calculated using Gaussian distributions.

This statistical analysis is repeated for the entire European domain and for the two major subdomains to investigate the heterogeneities of the temperature and equivalent temperature response to land cover transitions in the different climate regimes found in the analysis. Subdomain A mainly refers to central, southern and western Europe, and subdomain B to eastern Europe. The number of valid windows retrieved is 2779 for the analysis of the entire European domain, and 856 and 1032 for the subdomains A and B, respectively. Only the valid estimates for each land cover transition that occurred in at least 15% of the valid windows are considered as representative averages and shown in Fig. 6.

**Validation of model results**. We validated model outputs against relevant observation datasets. WRF model simulations based on the EURO-CORDEX configuration with the new land cover datasets LC1992 and LC2015 are compared with observation records for European climate (E-OBS)[93]. E-OBS data are averaged over the period 1992–2015 to align with the dataset used for the boundary conditions in model simulations. An additional simulation with the default IGBP land cover in WRF is added to benchmark the relative performance of the new ESA CCI land cover datasets with indices like the pattern correlation coefficient (PCC), the regional bias, and the root-mean-square error (RMSE). The three simulations show similar patterns for annual mean temperature relative to E-OBS, and the simulations with the CCI-based land cover datasets are found to reduce both model bias and RMSE (Supplementary Fig. 8).

Our simulations modeled multiple, and sometime contrasting, land cover changes per grid cell, and we could not directly compare the changes caused in the single component of the surface energy budget from specific LCCs with other available datasets. However, it was possible to validate estimates of surface albedo and soil moisture from the individual simulations. Estimates of surface albedo for LC1992 and LC2015 were resampled and compared against the satellite-derived CLARA_A2 dataset, which covers the 34-year period from 1982 until 2015 and provides monthly average values on a $0.25° \times 0.25°$ grid[94]. Monthly mean values (with standard deviation and pattern correlation) of 1992–2015 averaged surface albedo are compared in Supplementary Table 2. Results generally show high correlation coefficients and a consistent seasonal cycle, with values falling within the respective uncertainty ranges. The ESA CCI SM database[95] provides harmonized estimates of 10-cm soil moisture from a large set of satellite sensors

and the 1992–2015 average was used to compare soil moisture outputs from LC1992 and LC2015 (Supplementary Fig. 3). We found high pattern correlation coefficients (from 0.82 in summer to 0.96 in winter) and limited bias.

Temperature changes attributed to specific land transitions from our ridge-regression approach were compared to an alternative dataset based on satellite remote sensing observations[8,59]. This dataset is derived from potential vegetation changes where a space-for-time approximation is applied to multiscale remote sensing products and was developed to benchmark climate model outputs related to biophysical land processes. It includes changes in the surface energy balance and land surface temperature (day and night) for up to 10 IGBP land cover transitions (over affected land areas only) at a resolution of 1° for the period 2008–2012. We downloaded the publicly available version of this dataset[59] and postprocessed it to compute daily monthly mean surface temperature changes by a simple average between day and night temperature. The standard errors were derived from the corresponding standard deviations provided by the dataset. Because some of these estimates are likely unrealistic values (for example, >10 °C), we treat them as outliers and filter them out. We then compared the temperature effects of the individual land transitions with those in Fig. 6, for the EURO-CORDEX domain and for the subdomains A and B (Supplementary Table 1). We find that results are generally consistent among the two databases, but the comparison should consider some important caveats. First, care is needed when interpreting temperature data acquired using different protocols. The observational dataset reports about land surface temperature, i.e. the temperature at the top of the land surface, including bare land, water, snow or ice cover, cropland or forest canopy, while our study focuses on air (2 m) temperature. Although air temperature is recognized to be generally dependent on land surface temperature[61], there are inherent differences because satellite retrievals only occur under clear-sky conditions, and coupling strength between the two temperature indicators varies in time and across different land covers (e.g., coupling is stronger in forests than in open land)[62]. In addition, the observation dataset only refers to the LCCs between 2008 and 2012, a temporal dimension that differs from our study, and the reported temperature values for each specific LCC refer to the area affected by the land use change only, with the variables that are upscaled to one degree resolution. On the other hand, our study considers temperature values for a specific grid cell where the LCC is only a fraction of the area of the cell. Relatively smaller values are thus expected from our study.

**Reporting summary**. Further information on research design is available in the Nature Research Reporting Summary linked to this article.

## Data availability
Datasets of temperature changes and source data of their statistical decomposition are available as Supplementary Data file. Additional data are available from the corresponding author upon reasonable request.

## Code availability
The code (in R) of the ridge-regression analysis is available as Supplementary Data file.

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

## Acknowledgements

The support of the Norwegian Research Council is acknowledged through the projects Quiffin (project n. 254966), Bio4Fuels (project no. 257622), and BioPath (project no. 294534) by B.H. and F.C., and Bio4Clim (project no. 244074) and MitiStress (project no. 286773) by X.H. and F.C. W.Z. acknowledges support from the National Natural Science Foundation of China (project no. 41861134038). Simulations were performed on the resources provided by UNINETT Sigma2—the National Infrastructure for High Performance Computing and Data Storage in Norway.

## Author contributions

B.H., X.H., and F.C. designed the study. B.H. processed the land cover datasets, performed regional climate simulations, and postprocessed model outcomes with inputs from X.Z. X.H. and G.-A.F. performed the ridge-regression analysis. B.H. made the figures. B.H., X.H., X.Z., W.Z. and F.C. interpreted the results. F.C. wrote the paper with inputs from B.H., X.H., G.-A.F., X.Z., and W.Z.

## Competing interests

The authors declare no competing interests.
