## [Peer Review File · Nature Communications]

Review of “Prevailing local cooling from recent land cover changes in Europe”

This is an excellent paper and easy to review (although others need to examine their Bayesian regression approach as this is outside my area of expertise. I only have a few comments.

While the authors examine dry bulb temperature, in order to more completely assess the role of land cover change, they also should look at concurrent changes in absolute surface air humidity. The combined effect can be examined using surface air moist enthalpy: e.g.

Pielke Sr., R.A., C. Davey, and J. Morgan, 2004: Assessing "global warming" with surface heat content. *Eos*, 85, No. 21, 210-211.

Fall, S., N. Diffenbaugh, D. Niyogi, R.A. Pielke Sr., and G. Rochon, 2010: Temperature and equivalent temperature over the United States (1979 – 2005). *Int. J. Climatol.*, DOI: 10.1002/joc.2094.

There are other past papers that the authors should consider in their introduction to the first order effect of land use change/land management on weather and climate. These include, as just a few examples,

Pielke Sr., R.A., A. Pitman, D. Niyogi, R. Mahmood, C. McAlpine, F. Hossain, K. Goldewijk, U. Nair, R. Betts, S. Fall, M. Reichstein, P. Kabat, and N. de Noblet-Ducoudré, 2011: Land use/land cover changes and climate: Modeling analysis and observational evidence. *WIREs Clim Change* 2011, 2:828–850. doi: 10.1002/wcc.144.

Pielke Sr., R.A., R. Mahmood, and C. McAlpine, 2016: Land’s complex role in climate change. *Physics Today*, 69(11), 40.

Roy, S.S., R. Mahmood, D. Niyogi, M. Lei, S.A. Foster, K.G. Hubbard, E. Douglas, and R.A. Pielke Sr., 2007: Impacts of the agricultural Green Revolution - induced land use changes on air temperatures in India. *J. Geophys. Res. - Special Issue*, 112, D21108, doi:10.1029/2007JD008834

Strack, J.E., R.A. Pielke Sr, L.T. Steyaert, and R.G. Knox, 2008: Sensitivity of June near-surface temperatures and precipitation in the eastern United

States to historical land cover changes since European settlement. *Water Resources Research*, 44, W11401, doi:10.1029/2007WR00654

Zhang, T. X. Lin, R.A. Pielke Sr., and R. Mahmood, 2019: Irrigation impacts on surface air moist enthalpy in the central Great Plains of the USA. *Agriculture, Ecosystems and Environment*.

With respect to their abstract and text, they write “Land cover changes are an important driver of variations in local climate”. Yet the paper’s analysis is a regional study. They should change such text to “Land cover changes are an important driver of variations in local AND REGIONAL climate”.

As another example, the text “This study shows a quantification of how the mix of recent historical land cover changes in Europe can influence the local” should be “This study shows a quantification of how the mix of recent historical land cover changes in Europe can influence the local AND REGIONAL WEATHER AND CLIMATE “

The authors also need to add units; e.g. “prevailing average temperature cooling of -0.12 ± 0.20 ” should be “prevailing average temperature cooling of $-0.12 \text{ C} \pm 0.20 \text{ C}$ ”

With the authors considering my comments, I recommend acceptance.

Reviewer #2 (Remarks to the Author):

In "Prevailing local cooling from recent land cover changes in Europe" the authors use a recent and high resolution (300 m) land cover map provided through the ESA CCI as a boundary condition for a mesoscale climate model (WRF) to quantify the effect of land cover changes over Europe between 1992 and 2015, the period for which the land cover product is available. Although the aim of the study was not explicitly stated, the discussion hinted at "... informing about [individual land transitions] their local temperature effects ... aiming to assess biophysical effects from LCCs". Although the method appears valid, the interpretation of the model results would benefit from extensive model validation especially if the results are aiming to inform stakeholders about the biophysical effects from LCCs. Furthermore, both the model and the model experiment lack novelties. Hence, I'm afraid that even if the model results would pass all relevant validation tests, the study would still fail to move beyond the state-of-the-art. Even when anticipating methodological revisions, the current results appear more suitable for a specialist journal. The temperature effect of land transitions has been the topic of tens of model studies over the past decades. Almost every global model has run global and/or regional simulations to quantify the temperature effect of all sorts of afforestation and deforestation scenarios. Although most of these studies use slightly different land cover classes, the land cover classes used in this study are mostly in line with the previous studies. The current study differs from previous studies in using land cover data with the highest available spatial resolution this far. The simulation experiment, however, does not make use of the potential strength of the data. The 300-m resolution data were aggregated to 12 km, which is still high but exceeds the ~4 km threshold for precipitation resolving simulations. Contrary to the majority of the previous model studies, the mesoscale model could have been used to study the effects of sea breezes, mountain breezes, or precipitation. In my opinion the use of the high resolution land cover data and the mesoscale model did not translate into better research compared to that currently available. The idea of calculating the biophysical effect nicely complements ongoing effort to draw the attention to this omission from assessments of climate impacts. Nevertheless, owing to a lack of specific model developments or model parameterization there is no reason to assume that these results presented in this study are better than the results of any of the other models on the market. Hence, previous analyses can be used. If new simulations are made, as is the case in this study, I would expect an effort to move closer to results that can be used by stakeholders by accounting for the main tree species, the main crops, the change in standing biomass (relevant for roughness and albedo), the change in LAI (relevant for albedo and transpiration), soil moisture (relevant for transpiration), snow season (relevant for albedo)...

The study did not provide any model validation/evaluation results. Hence it is not clear whether the land surface model is up to the task? Ideally the coupled model should be evaluated; see for example Vanden Broucke et al. 2015 (10.1002/2015JD023095). If this is not possible or too demanding, the simulated temperature effects of land transitions should be tested against data and remote sensing products. If the model is used to disaggregate the observed temperature effect in its main driver, it should be demonstrated that the model simulates the correct changes in albedo, transpiration and roughness length following a land transition. Although CLM has an excellent track record, the model has not been extensively used over Europe. It would build confidence to see that CLM has an acceptable and realistic photosynthesis, transpiration, LAI, snow season, and soil water content over Europe as all affect the energy budget.

The finding of lower soil moisture in eastern Europe is new and should be confirmed by observations. This is counterintuitive as one would expect that the Mediterranean region of Europe is drier than the east. The heat bias in figure S8 largely coincides with the area for which cooling was observed which in turn matches the soil moisture map quite well. This is worrisome given the counterintuitive soil moisture distribution and needs independent testing. If soil moisture is confirmed to be the key driver, subdomains A&B should be replaced by a soil moisture gradient because the extent of dry soils is expected to increase, the current cooling effect of trees is thus expected to decrease as well (A & B are static but the real driver would then be spatially and temporally dynamic). This would lead to an interesting dilemma: should we plant trees in regions where they cool now - even when we don't need the extra cooling yet, or should we only encourage planting more trees in regions where they will cool the climate under future climate predictions?

Regression approaches to decompose a pixel-level signal into land cover specific signals have been used since long (see for example Pinty et al 2010, doi:10.1029/2010JD015372). The proposed

method may differ by the fact that a Bayesian framework rather than a maximum likelihood is used but I think that a statistical journal would be more suitable to publish that kind of progress.

Reviewer #1

This is an excellent paper and easy to review (although others need to examine their Bayesian regression approach as this is outside my area of expertise). I only have a few comments.

Thank you for the appreciation of our work.

While the authors examine dry bulb temperature, in order to more completely assess the role of land cover change, they also should look at concurrent changes in absolute surface air humidity. The combined effect can be examined using surface air moist enthalpy: e.g.

Pielke Sr., R.A., C. Davey, and J. Morgan, 2004: Assessing "global warming" with surface heat content. *Eos*, 85, No. 21, 210-211.

Fall, S., N. Diffenbaugh, D. Niyogi, R.A. Pielke Sr., and G. Rochon, 2010: Temperature and equivalent temperature over the United States (1979 - 2005). *Int. J. Climatol.*, DOI: 10.1002/joc.2094.

We investigated the effects of land cover changes to surface air humidity and used the new concept of equivalent temperature (\$T_E\$ ). There are now 3 new figures showing results using this indicator: Figure 3 shows the annual mean changes in surface humidity and equivalent temperature; Figure 5 shows the corresponding seasonality; Figure 6 also includes the decomposition of the effects on \$T_E\$ to the individual land transitions. Indeed, the use of this additional indicator provided complementary insights, that are presented and discussed in the paper (e.g., lines 133-151, 207-228, and 243-267).

There are other past papers that the authors should consider in their introduction to the first order effect of land use change/land management on weather and climate. These include, as just a few examples,

Pielke Sr., R.A., A. Pitman, D. Niyogi, R. Mahmood, C. McAlpine, F. Hossain, K. Goldewijk, U. Nair, R. Betts, S. Fall, M. Reichstein, P. Kabat, and N. de Noblet-Ducoudré, 2011: Land use/land cover changes and climate: Modeling analysis and observational evidence. *WIREs Clim Change* 2011, 2:828-850. doi: 10.1002/wcc.144.

Pielke Sr., R.A., R. Mahmood, and C. McAlpine, 2016: Land's complex role in climate change. *Physics Today*, 69(11), 40.

Roy, S.S., R. Mahmood, D. Niyogi, M. Lei, S.A. Foster, K.G. Hubbard, E. Douglas, and R.A. Pielke Sr., 2007: Impacts of the agricultural Green Revolution -induced land use changes on air temperatures in India. *J. Geophys. Res. - Special Issue*, 112, D21108, doi:10.1029/2007JD008834

Strack, J.E., R.A. Pielke Sr, L.T. Steyaert, and R.G. Knox, 2008: Sensitivity of June near-surface temperatures and precipitation in the eastern United States to historical land cover changes since European settlement. *Water Resources Research*, 44, W11401, doi:10.1029/2007WR00654

Zhang, T. X. Lin, R.A. Pielke Sr., and R. Mahmood, 2019: Irrigation impacts on surface air moist enthalpy in the central Great Plains of the USA. *Agriculture, Ecosystems and Environment*.

Thanks for pointing to these papers. We consulted them and included in the introduction and discussion of paper results as appropriate.

With respect to their abstract and text, they write “Land cover changes are an important driver of variations in local climate”. Yet the paper’s analysis is a regional study. They should change such text to “Land cover changes are an important driver of variations in local AND REGIONAL climate”. As another example, the text “This study shows a quantification of how the mix of recent historical land cover changes in Europe can influence the local” should be “This study shows a quantification of how the mix of recent historical land cover changes in Europe can influence the local AND REGIONAL WEATHER AND CLIMATE “.

This is now consistently changed throughout the revised manuscript.

The authors also need to add units; e.g. “prevailing average temperature cooling of -0.12 ± 0.20 ” should be “prevailing average temperature cooling of $-0.12 \text{ C} \pm 0.20 \text{ C}$ ”.

Units are now added.

With the authors considering my comments, I recommend acceptance.

Reviewer #2

In “Prevailing local cooling from recent land cover changes in Europe” the authors use a recent and high resolution (300 m) land cover map provided through the ESA CCI as a boundary condition for a mesoscale climate model (WRF) to quantify the effect of land cover changes over Europe between 1992 and 2015, the period for which the land cover product is available. Although the aim of the study was not explicitly stated, the discussion hinted at “... informing about [individual land transitions] their local temperature effects ... aiming to assess biophysical effects from LCCs”. Although the method appears valid, the interpretation of the model results would benefit from extensive model validation especially if the results are aiming to inform stakeholders about the biophysical effects from LCCs.

We increased the robustness of our analysis by validating results and key variables against observational datasets. We compared the final decomposed temperature signal to the individual land cover transitions with a recently available observationally-driven dataset for similar land cover changes (*lines 268-298 and Supplementary Table S1*). Results are found to be broadly consistent, including the contrasting response to tree cover between the western and eastern part of Europe. It was also possible to compare key variables like surface albedo and soil moisture between our simulations and observational datasets (*Supplementary Figure S3 and Table S2*). We also compared our WRF model simulations based on the new land cover datasets and the EURO-CORDEX configuration with observation records for European climate (*Supplementary Figure S8*), showing that we achieved better performances than the default IGBP land cover in WRF. Overall, we did not find critical flaws that can question the validity of our main conclusions. In the revised text, we expanded the discussion of the comparison of our results with the existing literature (*lines 152-173, 268-298, and 318-343*). We also added a new subsection at the end of the Methods where we describe all the datasets and approaches used in the comparison of our results with the literature (*lines 596-649*).

Regarding the sentence reported in the comment. As this is the first studies to estimate temperature effects at a continental level from the mix of historical recent land transitions in Europe, it was not intended to produce all-inclusive results for direct application to all stakeholders, but to provide novel insights for the scientific community in general. We revised the sentence to prevent misunderstanding. We also better explained the aim of our analysis at the end of the introduction (*lines 69-79*) and at the beginning of the discussion (*lines 301-317*).

Furthermore, both the model and the model experiment lack novelties. Hence, I'm afraid that even if the model results would pass all relevant validation tests, the study would still fail to move beyond the state-of-the-art. Even when anticipating methodological revisions, the current results appear more suitable for a specialist

journal. The temperature effect of land transitions has been the topic of tens of model studies over the past decades. Almost every global model has run global and/or regional simulations to quantify the temperature effect of all sorts of afforestation and deforestation scenario's. Although most of these studies use slightly different land cover classes, the land cover classes used in this study are mostly in line with the previous studies.

To the best of our knowledge, we are not aware of any other study that quantified or investigated the regional climate response to recent historical land cover changes in Europe at a continental level. We tried to better explain these novel aspects in the revised paper, both at the end of the introduction (*lines 69-79*) and at the beginning of the discussion (*lines 301-317*). Following the suggestions of Reviewer #1, we also expanded the results and included effects on surface humidity and equivalent temperature, thereby adding additional elements of novelty to the study. We explicitly point to the fact that most of the previous studies focused on only one individual large-scale land cover transition at a time to explore the specific regional climate effects. Here, we simultaneously consider a variety of historical LCCs per grid cell and investigate both the net effects at a continental level and the contributions from each individual land transition using a Bayesian approach that can directly connect changes in temperature to the specific LCCs.

The current study differs from previous studies in using land cover data with the highest available spatial resolution this far. The simulation experiment, however, does not make use of the potential strength of the data. The 300-m resolution data were aggregated to 12 km, which is still high but exceeds the ~4 km threshold for precipitation resolving simulations. Contrary to the majority of the previous model studies, the mesoscale model could have been used to study the effects of sea breezes, mountain breezes, or precipitation. In my opinion the use of the high resolution land cover data and the mesoscale model did not translate into better research compared to that currently available.

The resolution of 0.11 degree (EUR-11, ~12 km) is the highest available from the CORDEX regional climate model simulations for the European domain (EURO-CORDEX). By adopting the same resolution in our study, we can use the parameterization and model configuration of WRF available from the EURO-CORDEX database, which was developed and calibrated after validation of the spatiotemporal patterns of the European climate against observational references. This specificity is now added in the Methods (*lines 465-466*). Further, the high resolution information of the land cover database is retained in our experiments. Per each grid cell, the energy balance and surface fluxes are calculated for each fraction of land types within the grid cell, and then aggregated at the grid scale based on the respective proportions. In general, we had to adapt the resolution of the experiments to the objectives of our study, and to the available calibrated parameterizations.

Assessing effects on sea breezes, mountain breezes, etc., is beyond the scope of our analysis.

The idea of calculating the biophysical effect nicely complements ongoing effort to draw the attention to this omission from assessments of climate impacts. Nevertheless, owing to a lack of specific model developments or model parameterization there is no reason to assume that these results presented in this study are better than the results of any of the other models on the market. Hence, previous analyses can be used. If new simulations are made, as is the case in this study, I would expect an effort to move closer to results that can be used by stakeholder by accounting for the main tree species, the main crops, the change in standing biomass (relevant for roughness and albedo), the change in LAI (relevant for albedo and transpiration), soil moisture (relevant for transpiration), snow season (relevant for albedo)...

The aim of our study is an application of a state-of-the-art model to a research question that was not investigated before. We updated the land cover dataset of the model with a new observationally derived product with improved representation of land cover classes. We think that the regional climate effects at a continental scale of recent land cover changes in Europe could not be inferred from previous analysis. We isolated the main drivers of regional climate change, which were discussed and compared with existing knowledge from the scientific literature. Further primary model developments are beyond our objectives. It is not uncommon to publish studies where a state-of-the-art climate model is used for a specific research question, to mention a few: Naudts et al. (2016) *Science* 351:597-600; Luysaert et al. (2018) *Nature* 562:259-262; Findell et al. (2017) *Nature Communications* 8:989; Seneviratne et al. (2018) *Nature Geoscience* 11:88-96; Davin et al. (2014) *PNAS* 111:9757-9761. Overall, we agree that future availability of high-resolution maps of gradients in vegetation properties between (and within) land cover classes, together with improvements in representation of plant phenology, can offer opportunities to increase model capabilities. We refer to this potential and to the possible limitations related to the use of a single model in the discussion section (*lines 318, 343, 344-356 and 375-382*). We also explicitly argue that our analysis could be instrumental to stimulate additional research that can ultimately lead to multi-model inter-comparison efforts to study the effects of recent LCCs on European climate (*lines 338-343*).

The study did not provide any model validation/evaluation results. Hence it is not clear whether the land surface model is up to the task? Ideally the coupled model should be evaluated; see for example Vanden Broucke et al. 2015 (10.1002/2015JD023095). If this is not possible or too demanding, the simulated temperature effects of land transitions should be tested against data and remote

sensing products. If the model is used to disaggregate the observed temperature effect in its main driver, it should be demonstrated that the model simulates the correct changes in albedo, transpiration and roughness length following a land transition. Although CLM has an excellent track record, the model has not been extensively used over Europe. It would built confidence to see that CLM has an acceptable and realistic photosynthesis, transpiration, LAI, snow season, and soil water content over Europe as all affect the energy budget.

We invested additional efforts to validate our results and increase their robustness. We tested the implementation of the new land cover dataset in the coupled WRF-CLM model against observations (E-OBS) and we found that statistical scores are improved relative to the previous dataset (*Figure S8*). Given the setup of our study (based on a mix of simultaneous land cover changes per grid cell instead of one large-scale simulation per individual transition), we could not directly infer the changes in the single variables of the surface energy budget for each LCC for comparison with other studies. However, we now tested key variables, such as soil moisture and surface albedo, produced with the new land datasets against observational records, and we compared our final temperature effects per LCC with satellite-retrievals. Monthly mean values and spatial patterns of surface albedo estimates were compared to the satellite-derived CLARA-A2 dataset (*Supplementary Table S2*). Soil moisture seasonality and spatial patterns were compared with the European Space Agency Soil Moisture database (*Supplementary Figure S3*). The decomposition of the temperature signal to individual land cover transitions was compared to a semi-empirical database that collected land surface temperature changes following LCCs (*Supplementary Table S1*). We discussed these comparisons in the paper and added a new subsection to the Methods (*lines 596-649*). Overall, these validation efforts did not show relevant divergences of our model results with observations, and we believe they helped to improve the robustness of our findings.

The finding of lower soil moisture in eastern Europe is new and should be confirmed by observations. This is counterintuitive as one would expect that the Mediterranean region of Europe is drier than the east. The heat bias in figure S8 largely coincides with the area for which cooling was observed which in turn matches the soil moisture map quite well. This is worrisome given the counterintuitive soil moisture distribution and needs independent testing. If soil moisture is confirmed to be the key driver, subdomains A&B should be replaced by a soil moisture gradient because the extent of dry soils is expected to increase, the current cooling effect of trees is thus expected to decrease as well (A & B are static but the real driver would then be spatially and temporally dynamic). This would lead to an interesting dilemma: should we plant trees in regions where they cool now - even when we don't need the extra cooling yet, or should we only encourage planting more trees in regions where they will cool the climate under future climate predictions?

We agree that the role of soil moisture is important, although it is not the sole driver, and we better elaborated on the associated mechanisms. The average lower soil moisture content of eastern Europe is discussed in previous studies as well, and we now include them in the paper (*lines 160-173*). As mentioned above, we validated the soil moisture results with the European Space Agency Soil Moisture database which provides harmonized estimates of soil moisture from a large set of satellite sensors. We found good statistical scores (*Supplementary Figure S3*). We considerably expanded the discussion on the role of soil moisture. We directly refer to the mechanism of the soil moisture temperature feedbacks in climatic transition areas (such as the one discussed in our paper, which involves the transition from oceanic to continental climate), and used a variety of specific new references to support and explain these aspects (*lines 155-173*). The definition of our subdomains largely reflects average soil moisture content. The influence of the bias mentioned in the comment above is reduced because our study assesses the difference between two simulations (LC2015 and LC1992), and since lateral boundary conditions do not vary between experiments, absolute biases tend to cancel out and the differences in model outcomes can be attributed to the different land cover only.

The last point of the comment indeed offers an interesting consideration. We performed our analysis under present day climate conditions (averaged 1992-2015) to assess the effects of recent LCCs. It has been shown that effects of LCCs on regional climate are highly dependent on background conditions (Pitman (2011) *Nature Clim. Change* 1:472-475), and our results can thus change under a future warmer and dryer climate. We now refer to this aspect in the discussion section of the paper (*lines 344-356*).

Regression approaches to decompose a pixel-level signal into land cover specific signals have been used since long (see for example Pinty et al 2010, doi:10.1029/2010JD015372). The proposed method may differ by the fact that a Bayesian framework rather than a maximum likelihood is used but I think that a statistical journal would be more suitable to publish that kind of progress.

We consulted the paper and, as far as we can see, we did not find significant risks of similarities with our study (neither for the thematic application area nor for the analytical approach used). We are not aware of previous modelling studies that directly inferred temperature changes for each LCC from simulations based on mixed land transitions. Common practice in previous studies was to perform one area-extended simulation per LCC and/or rely on the reassembly of the individual components in the surface energy balance. We better highlighted these aspects in the revised paper (*lines 231-239, 308-317*).

Reviewers' comments:

Reviewer #1 (Remarks to the Author):

The response to the reviews is thorough and persuasive. I recommend acceptance. My only further suggestion, is that since text was included on policy implications, that these other papers be considered:

Marland, G., R.A. Pielke, Sr., M. Apps, R. Avissar, R.A. Betts, K.J. Davis, P.C. Frumhoff, S.T. Jackson, L. Joyce, P. Kauppi, J. Katzenberger, K.G. MacDicken, R. Neilson, J.O. Niles, D. Dutta S. Niyogi, R.J. Norby, N. Pena, N. Sampson, and Y. Xue, 2003: The climatic impacts of land surface change and carbon management, and the implications for climate-change mitigation policy. *Climate Policy*, 3, 149-157.

Mahmood, R., R.A. Pielke Sr., T.R. Loveland, and C.A. McAlpine, 2016: Climate relevant land use and land cover change policies. *Bull. Amer. Meteor. Soc.* 195-202,

Roger A. Pielke Sr

Reviewer #2 (Remarks to the Author):

The revisions carefully considered all technical concerns and confirmed the earlier assessment that the methods appear valid. From a technical point of view this is a very good study and the manuscript is easy to read. Nevertheless, the revisions circumvented the more scientific concerns of one of the reviewers with the classic "out of scope" argument. The authors are of course free to choose the scope of their study but given their choice it is not convincing that the study answers a novel and meaningful question. Although there are some technical advancements, I don't think this is a key study that will force its readers to think differently about land cover change. I agree with the authors that this may be the first study to use a(n aggregated) high resolution land cover change (LCC) map in combination with a mesoscale atmospheric model and a statistical disaggregation method to attribute temperature changes to specific land cover transitions but the finding that the transition from agricultural land to forests plays a central role in the energy budget of the land surface has already been demonstrated by, for example, Alkame and Cescatti 2016 (their fig 3).

The authors present the statistical disaggregation method as one of the key novelties. The method might be very useful to disaggregate observed mixed signals but the study under review is a modelling study in which factorial experiments could have been used (and are used, i.e., the no crop simulation) to disaggregate the simulated temperature effect. As the disaggregation method seems valid, the authors can use the method they prefer but the statistical method is unlikely to be essential to obtain the results presented in the study.

Some key aspects of the study, i.e., the time period considered, are poorly justified. The authors limit the length of their study to the availability of remote sensing data. This is an understandable choice but not necessarily a scientifically interesting one. Did something happen in that period that makes it worth to have a closer look at these years? Can the LCC be linked to a policy or even a tendency/fashion in land management? The risk of analyzing short term data from slowly maturing ecosystems (= forests) is that one jumps to conclusions too quickly. An unanswered key question is whether the initial cooling effect resulting from the transition of a cropland to a forest will be maintained when the forest ages?

The title is misleading because the question whether LCC resulted in a cooling or a warming can only be answered by considering both the biogeochemical (C-cycle) and biophysical (this study) effect. It should be clear from reading the title that the C-cycle was omitted from this study (or that the sole focus of the study is on the biophysical effects).

The abstract is superficial. The -0.1 and +0.1 changes are reported but the logic of the idealized simulations is not explained neither are the consequences of this finding. The last sentence is too vague and lacks nuance. It now implies that we should consider abandoning all croplands if that results in a cooling (remember the discussion following the publication by Bala et al 2007). Better articulate the implications of your study for European land use and possibly resulting land cover changes.

L149 "manly" -> "mainly"

L212 "means" -> "mean"

L289 "are closer to observations than modelling studies" -> "are closer to observations than other modelling studies"

L293 "The response is expected to be stronger for large-scale transitions, because they can trigger more substantial direct and indirect effects." -> I don't think this is true. The effects of large-scale transitions will take longer (vertically and spatially for the temperature and moisture signals) to mix, so it becomes easier to detect but the effect should be the same. The same cooling per unit of change can be expected irrespective of whether 1m² or 1km² of land is changed. If you think this is not correct, then use primary literature sources to make your point.

L308 "climatological albedo" What is the climatological albedo? How does it differ from the "albedo" used in the rest of the manuscript?

L341 "Additional research is also required to improve representation of plant physiology and vegetation dynamics to increase the accuracy of model outcomes." -> be more specific. The current phrasing is handwaving. It is not clear which findings supported this conclusion in the first place. Does this imply that the remote sensing based LCC maps and the atmospheric model WRF need no further improvements to increase the accuracy of model outcomes?

Reviewer #3 (Remarks to the Author):

1. Actually, the Bayesian linear regression used in this study can be thought be as a generalized linear model (GLM) with a Gaussian distribution. In the GLM, an assumption of a Gaussian distribution for the error term epsilon should be confirmed.
2. Authors considered a multivariate Gaussian distribution (MVN) with a mean value of zero and covariance matrix for a prior distribution beta. In this setting, one can use a conjugate distribution of the MVN which is the normal-inverse Wishart distribution with degrees of freedom (ν) and scale matrix (Ψ) as the conjugate prior for the covariance matrix of the MVN. However, there are no explanations on the prior distribution for the covariance matrix of MVN.
3. Authors need to provide posterior distributions for all the parameters so that we can evaluate the significance of parameters in the GLM model with uncertainty range (or credible interval) in the context of the p-value. For example, we can conclude that the estimated parameters are significant if the uncertainty range of the parameters is relatively narrow and their signs are consistent over the entire range of posterior distribution. The significance of the parameters should be first examined before the evaluation of the difference in regression parameters.
4. In Figure 2, the label PDF (%) at y-axis appears quite awkward to me. It should be just a probability ($f(x)$).
5. Author needs to include the null-hypothesis significance test for the annual average temperature changes.

Reviewer #1:

The response to the reviews is thorough and persuasive. I recommend acceptance. My only further suggestion, is that since text was included on policy implications, that these other papers be considered:

Marland, G., R.A. Pielke, Sr., M. Apps, R. Avissar, R.A. Betts, K.J. Davis, P.C. Frumhoff, S.T. Jackson, L. Joyce, P. Kauppi, J. Katzenberger, K.G. MacDicken, R. Neilson, J.O. Niles, D. Dutta S. Niyogi, R.J. Norby, N. Pena, N. Sampson, and Y. Xue, 2003: The climatic impacts of land surface change and carbon management, and the implications for climate-change mitigation policy. *Climate Policy*, 3, 149-157.

Mahmood, R., R.A. Pielke Sr., T.R. Loveland, and C.A. McAlpine, 2016: Climate relevant land use and land cover change policies. *Bull. Amer. Meteor. Soc.* 195-202,

Roger A. Pielke Sr

Thanks for the appreciation of our work. We consulted these two references and included them in the final section of the manuscript.

Reviewer #2:

The revisions carefully considered all technical concerns and confirmed the earlier assessment that the methods appear valid. From a technical point of view this is a very good study and the manuscript is easy to read. Nevertheless, the revisions circumvented the more scientific concerns of one of the reviewers with the classic "out of scope" argument. The authors are of course free to choose the scope of their study but given their choice it is not convincing that the study answers a novel and meaningful question. Although there are some technical advancements, I don't think this is a key study that will force its readers to - think differently about land cover change. I agree with the authors that this may be the first study to use a(n aggregated) high resolution land cover change (LCC) map in combination with a mesoscale atmospheric model and a statistical disaggregation method to attribute temperature changes to specific land cover transitions but the finding that the transition from agricultural land to forests plays a central role in the energy budget of the land surface has already been demonstrated by, for example, Alkame and Cescatti 2016 (their fig 3).

Our aim was to estimate the temperature effects from recent historical land cover changes at a European level, which to the best of our knowledge were never explored before. We achieved this goal by integrating multiple elements within a consistent interdisciplinary

approach. Alkama and Cescatti (2016) did not assess regional climate effects from land cover changes, but recent forest losses and gains using remotely sensed retrievals only.

The authors present the statistical disaggregation method as one of the key novelties. The method might be very useful to disaggregate observed mixed signals but the study under review is a modelling study in which factorial experiments could have been used (and are used, i.e., the no crop simulation) to disaggregate the simulated temperature effect. As the disaggregation method seems valid, the authors can use the method they prefer but the statistical method is unlikely to be essential to obtain the results presented in the study.

The benefits of the statistical disaggregation are better explained in the revised manuscript (Lines 70-80). Running factorial experiments for each single LCC is usually very resource and time intensive. The statistical method proposed in our study allows to single out the temperature effect of each of the 10+ land cover changes (LCC) from a single realistic simulation where multiple LCCs per grid cell occur at the same time.

Some key aspects of the study, i.e., the time period considered, are poorly justified. The authors limit the length of their study to the availability of remote sensing data. This is an understandable choice but not necessarily a scientifically interesting one. Did something happen in that period that makes it worth to have a closer look at these years? Can the LCC be linked to a policy or even a tendency/fashion in land management? The risk of analyzing short term data from slowly maturing ecosystems (= forests) is that one jumps to conclusions too quickly. An unanswered key question is whether the initial cooling effect resulting from the transition of a cropland to a forest will be maintained when the forest ages?

The time period of our analysis is determined by the availability of the timeseries of the land cover data (from 1992 to 2015). This represents a relatively long time period with about 70 Mha of LCCs occurred in Europe. We tried to connect these LCCs with some major trends in socio-economic context in Europe such as urbanization and declines in agricultural activities, or climate change (lines 88-105). We also refer to other studies that discussed the trends and causes of land use changes in Europe in more details (lines 97-98). Regarding vegetation dynamics, we agree that their representation has potential for improvements in both remotely sensed products and climate models. We added a specific discussion on the topic to the manuscript (lines 342-350).

The title is misleading because the question whether LCC resulted in a cooling or a warming can only be answered by considering both the biogeochemical (C-cycle) and biophysical (this study) effect. It should be clear from reading the title that the C-cycle was omitted from this study (or that the sole focus of the study is on the biophysical effects).

The word "biophysical" is added to the title.

The abstract is superficial. The -0.1 and +0.1 changes are reported but the logic of the idealized simulations is not explained neither are the consequences of this finding. The last sentence is too vague and lacks nuance. It now implies that we should consider abandoning all croplands if that results in a cooling (remember the discussion following the publication by Bala et al 2007). Better articulate the implications of your study for European land use and possibly resulting land cover changes.

We revised the abstract by adding a sentence about the seasonal effects. Due to strict word limitations (150 words), we could not accommodate further explanations about the methods (which will inevitably cause deletion of text somewhere else). We also changed the last sentence and made it more grounded on the implications of our study and on the importance of the specific regional context.

L149 “manly” -> “mainly”

Changed.

L212 “means” -> “mean”

Changed.

L289 “are closer to observations than modelling studies” -> “are closer to observations than other modelling studies”

Changed.

L293 “The response is expected to be stronger for large-scale transitions, because they can trigger more substantial direct and indirect effects.” -> I don’t think this is true. The effects of large-scale transitions will take longer (vertically and spatially for the temperature and moisture signals) to mix, so it becomes easier to detect but the effect should be the same. The same cooling per unit of change can be expected irrespective of whether 1m² or 1km² of land is changed. If you think this is not correct, then use primary literature sources to make your point.

We now added scientific references to better support this statement (lines 294-295). For example: Li et al. showed that, compared with regional deforestation, global deforestation is greatly amplified especially in temperate and boreal regions (<https://link.springer.com/article/10.1007/s00704-010-0302-y>); Lombardi et al. show a non-linear increase in the temperature response to increasing fractions of forest removals (from 5 to 100%) (<https://doi.org/10.5194/bgd-9-14639-2012>). More generally, the review by Perugini et al. find that larger-scale LCCs trigger large feedbacks through e.g. ocean and sea-ice dynamics that may amplify climate impacts. Area-limited LCCs mostly have local effects at the surface and first order interactions with the boundary layer, but not large-scale feedbacks (<https://doi.org/10.1088/1748-9326/aa6b3f>).

L308 “climatological albedo” What is the climatological albedo? How does it differ from the “albedo” used in the rest of the manuscript?

Climatological albedo refers to albedo values including climatic conditions (snow, humidity). We now simply call it surface albedo as in the rest of the manuscript.

L341 “Additional research is also required to improve representation of plant physiology and vegetation dynamics to increase the accuracy of model outcomes.” -> be more specific. The current phrasing is handwaving. It is not clear which findings supported this conclusion in the first place. Does this imply that the remote sensing based LCC maps and the atmospheric model WRF need no further improvements to increase the accuracy of model outcomes?

This is now better explained in the paper. Following the input from a Reviewer’s comment above, we specifically make the example of improvements for vegetation dynamics in both remotely sensed products of land cover classes and regional climate models (Lines 342-350).

Reviewer #3:

Actually, the Bayesian linear regression used in this study can be thought be as a generalized linear model (GLM) with a Gaussian distribution. In the GLM, an assumption of a Gaussian distribution for the error term epsilon should be confirmed.

We calculated the residuals $\hat{\varepsilon} = y - x\hat{\beta}$ for all the regressions and their mean and median are close to zero. The histogram of the residuals is unimodal and very close to symmetric, and similar to the Gaussian distribution. Further, the distribution of the differences in the regression coefficients will be even closer to Gaussian than the residuals due to the central limit theorem. We have added these additional explanations to the methods (line 594-596).

Authors considered a multivariate Gaussian distribution (MVN) with a mean value of zero and covariance matrix for a prior distribution beta. In this setting, one can use a conjugate distribution of the MVN which is the normal-inverse Wishart distribution with degrees of freedom (ν) and scale matrix (Ψ) as the conjugate prior for the covariance matrix of the MVN. However, there are no explanations on the prior distribution for the covariance matrix of MVN.

We agree that for a multivariate Gaussian distribution, one could use a conjugate distribution for the covariance matrix. The key reasons for choosing a fixed covariance matrix in $N(0, \sigma_\varepsilon^2 I)$ with σ_ε^2 large is that this prior is proper, almost shift-invariant and non-informative. It is widely used in statistics because it is easy to implement (since it only requires one hyperparameter) and it only imposes weak influence on the results in terms of minor stabilization when the data do not inform about a regression coefficient.

A normal-inverse Wishart has many parameters for the covariance matrix Ψ of the regression coefficients. In our setting, we only have 25 datapoints (given by the 5 by 5 grids in the moving window) and 16 coefficients (given by the land cover classes) in each window. This means that it is not plausible to learn about covariance structure of the regression coefficients, and the choice of the scale matrix in the inverse-Wishart distribution may have undue influence on the inference for the regression coefficients. Additionally, the regression is performed more than two thousand times, and ensuring that the degrees of freedom and the scale matrix chosen for the inverse-Wishart distribution is robust while not unduly influencing the results is a complex task. We therefore considered the simple prior $N(0, \sigma_\epsilon^2 I)$ a better choice in the setting of our experiments, as it is almost translation-invariant and non-informative, and no prior expert knowledge is required. We have added this additional explanation to the paper (lines 545-549).

Authors need to provide posterior distributions for all the parameters so that we can evaluate the significance of parameters in the GLM model with uncertainty range (or credible interval) in the context of the p-value. For example, we can conclude that the estimated parameters are significant if the uncertainty range of the parameters is relatively narrow and their signs are consistent over the entire range of posterior distribution. The significance of the parameters should be first examined before the evaluation of the difference in regression parameters.

In our setting, the individual regression coefficients do not have marginal interpretations. One land cover must always transition to another since $\sum_j X_{i,j} = 0$ for the covariate values (lines 526-529). This is the reason why we do not consider posteriors of the individual regression coefficients, and only consider posteriors for the differences in the regression coefficients. In the context of Bayesian statistics, the posterior of the difference between two regression coefficients can be informative even when the data do not contain information about the two regression coefficients separately. When the data do not inform about the difference between two regression coefficients this manifests a large posterior variance.

The posterior for the transition $\Delta y_{a \rightarrow b}$ at each square is described by a mean and a standard deviation. As argued by the Reviewer, this posterior typically will not suggest that the effect of the transition is significant for any (5 x 5) square since the data in a single square do not contain enough information to robustly inform about the transition. This is why we pool information across all the squares and then consider the posterior of the average effect, which has a posterior mean and variance that can be calculated from the individual posterior means and variances from the different squares (lines 580-593). These average effects of land transitions on temperature are the quantities of interest in our study.

In Figure 2, the label PDF (%) at y-axis appears quite awkward to me. It should be just a probability (f(x)).

This is now changed in the revised paper.

Author needs to include the null-hypothesis significance test for the annual average temperature changes.

The main contribution of the statistical method is to disaggregate temperature effects from climate model outputs to the individual land transitions. Climate model outputs were compared against observations, as well as the disaggregated temperature effects from the statistical analysis (lines 607-661).

Reviewers' comments:

Reviewer #4 (Remarks to the Author):

The attached review comments about the statistical analysis. I find that the communication of the model is poor. As a statistician, I strongly believe that the model must be presented clearly and correctly -- just as a discipline scientist would expect my scientific interpretations to be correct when collaborating. With changes to the model statement to correct errors and allow for reproducibility in the analysis, the manuscript would be improved.

Review: Prevailing regional biophysical cooling from recent land cover changes in Europe

In what follows, I address the statistical analyses presented in the manuscript. Overall the manuscript is well written; however, I have concerns about the model presentation – I suspect that when the model is properly defined the issues presented below will be resolved as the rest of the manuscript is reasonably well written and clear. However, the language used in the manuscript to describe the statistical model does not make the model clear and this causes issues. What follows might sound somewhat negative at time – this review is meant to be constructive criticism to make sure the analyses and language you use represents the actual model and inference that you are presenting.

First, this is not a Bayesian model – it is a frequentist ridge regression model. Ridge regression has a Bayesian interpretation as a prior on β , however, Bayesian estimates don't wear hats – Bayesian estimates are inherently distributions, not point estimates. For this model, you have the analytic distribution – there is not a need for fitting the model using Markov Chain Monte Carlo – however, the estimation process presented is frequentist and definitely **not Bayesian**. In fact, you even use the phrase “least-square estimate” before equation (6) – this is a frequentist estimate. Please remove the references to Bayesian models if you aren't actually doing Bayesian inference. This is an incorrect and misleading statement.

There are a number of specific notational and modeling issues not addressed in the model statement that need to be resolved, otherwise from a statistical perspective, the analyses presented in the paper cannot be reproduced given the model statement.

- The model statement is very unclear – this would not be an acceptable model statement in an advanced undergraduate or introductory graduate applied statistics course. Please clean up the notation because if this was a published article I, as a statistician reader, would have severe doubts about the statistical validity of the results presented in the paper just as you would have about the scientific impact if I were writing a statistics paper in which I mis-spoke about fundamental scientific terms, definitions, and interpretations.
- Are different β s estimated for each window? If so, shouldn't these be indexed by site i (and also by a or b) and how would this model be identifiable? If the β s are constant across windows, this needs to be made clear. As currently written, the model in equations (1)-(7) implies only one β for all windows $i = 1, \dots, N$ and all time periods a and b . I doubt this is the model you actually fit so change the notation to reflect this.
- What was the choice of σ_β^2 used as a prior? What is the effect of re-fitting the analysis with different values of σ_β^2 ? I suspect the results are highly sensitive to the choice of σ_β^2 indicating that the results are sensitive to the prior.
- What are the meaning of β_a and β_b in equation (2)? Do the X or y variables also change with a and b ? I don't follow the notation. How is there no index of a or b in equations (1)-(7) but it shows up out of the blue in equation (8)? How do I interpret equations (1)-(7) with respect to the a and b ? Are each of these equations fit for the a and b groups? Are the a and b groups the different time periods? Be clear in what you mean – as a reader this lack of attention to detail will cause me to doubt the entire analysis.
- Equation (9) – How do you estimate $\hat{\sigma}_{ab}$ under the least-squares frequentist estimate?
- Equation (10) needs an identity matrix to be correct: $C = \left(\mathbf{X}'\mathbf{X} + \frac{\sigma_\varepsilon^2}{\sigma_\beta^2} \mathbf{I} \right)^{-1} \sigma_\varepsilon^2$
- Equation (11): what are $\hat{\mu}_i$ and $\hat{\sigma}_i$? These are not defined anywhere. Please define these.
- You also state that windows where $\hat{\sigma}_i$ is uninformative are omitted are dropped (lines 598-605) – what does this mean? Just because you think the results are “too uncertain” doesn't mean that they are

invalid. This must over-inflate your confidence in results and causes me to question the results presented. Dropping results that are “uninformative” is **definitely not a Bayesian** thing to do.

- Figure 6: If you have a *Bayesian* distribution, you would plot the distribution, not a bar with error bars. The bar plot of a mean with error bars is one of the worst plots – you have so many nice graphics and plots – why not use a box and whiskers plot or a violin plot instead? These are the common way of displaying Bayesian estimates because they convey that the estimate is a distribution, not a point estimate.

Reviewer #4:

In what follows, I address the statistical analyses presented in the manuscript. Overall the manuscript is well written; however, I have concerns about the model presentation – I suspect that when the model is properly defined the issues presented below will be resolved as the rest of the manuscript is reasonably well written and clear. However, the language used in the manuscript to describe the statistical model does not make the model clear and this causes issues. What follows might sound somewhat negative at time – this review is meant to be constructive criticism to make sure the analyses and language you use represents the actual model and inference that you are presenting.

Answer: Thank you for the overall appreciation of our work, and for the specific suggestions to improve the clarity of the statistical part of the methods. We took all the inputs below into careful consideration while revising the corresponding Methods section (Line: 498-592).

First, this is not a Bayesian model – it is a frequentist ridge regression model. Ridge regression has a Bayesian interpretation as a prior on β , however, Bayesian estimates don't wear hats – Bayesian estimates are inherently distributions, not point estimates. For this model, you have the analytic distribution – there is not a need for fitting the model using Markov Chain Monte Carlo – however, the estimation process presented is frequentist and definitely not Bayesian. In fact, you even use the phrase “least-square estimate” before equation (6) – this is a frequentist estimate. Please remove the references to Bayesian models if you aren't actually doing Bayesian inference. This is an incorrect and misleading statement.

Answer: We agree that the estimation methods are of a frequentist ridge regression nature and we rephrased the text accordingly. We enjoy the Bayesian framework for model construction, but we acknowledge that presenting the full analysis as a Bayesian interpretation can be misleading. We have now updated the manuscript and the references accordingly (Line: 17, 75, 232-233, 314, etc.).

There are a number of specific notational and modeling issues not addressed in the model statement that need to be resolved, otherwise from a statistical perspective, the analyses presented in the paper cannot be reproduced given the model statement.

- The model statement is very unclear – this would not be an acceptable model statement in an advanced undergraduate or introductory graduate applied statistics course. Please clean up the notation because if this was a published article I, as a statistician reader, would have severe doubts about the statistical validity of the results presented in the paper just as you would have about the scientific impact if I were writing a statistics paper in which I mis-spoke about fundamental scientific terms, definitions, and interpretations.

Answer: We have largely rewritten the model statement and the corresponding section presenting the ridge-regression-based method, improved the clarity of the notation, and took into account all comments below (Line: 510-584).

- Are different β_s estimated for each window? If so, shouldn't these be indexed by site i (and also by a or b) and how would this model be identifiable? If the β_s are constant across windows, this needs to be made clear. As currently written, the model in equations (1)-(7) implies only one β for all windows $i = 1, \dots, N$ and all time periods a and b . I doubt this is the model you actually fit so change the notation to reflect this.

Answer: Different β_s are estimated for each window. They are now indexed by window i . We also changed the notation (a and b are now j and k), and their meaning is now clearly explained in the corresponding section of the methods (line: 528-530). The revised text now clearly states how the models are estimated (lines: 531-546), and how the results for the windows are combined (Line: 570-584).

- What was the choice of σ_B^2 used as a prior? What is the effect of re-fitting the analysis with different values of σ_B^2 ? I suspect the results are highly sensitive to the choice of σ_B^2 indicating that the results are sensitive to the prior.

Answer: Due to the change to ridge regression, β_i no longer has a prior. We investigated the sensitivity to the

penalty parameter λ in the ridge regressions, and we found minimal changes by changing λ from 10^{-12} to 10^{-11} or 10^{-13} (Line: 540-541).

- What are the meaning of β_a and β_b in equation (2)? Do the X or y variables also change with a and b? I don't follow the notation. How is there no index of a or b in equations (1)-(7) but it shows up out of the blue in equation (8)? How do I interpret equations (1)-(7) with respect to the a and b? Are each of these equations fit for the a and b groups? Are the a and b groups the different time periods? Be clear in what you mean – as a reader this lack of attention to detail will cause me to doubt the entire analysis.

Answer: The *a* and *b* referred to different land covers so that $a \rightarrow b$ denoted the transition from land cover *a* to land cover *b*. We see the potential for confusion in introducing these symbols, and we have rewritten this part completely when changing the presentation from Bayesian inference to ridge-based regression (Line: 531-546).

- Equation (9) – How do you estimate $\hat{\sigma}_{ab}$ under the least-squares frequentist estimate?

Answer: We estimate the residual variance, then the covariance matrix of the coefficients, and apply this to find the estimate of the transition from land cover *j* to land cover *k*, $\beta_k - \beta_j$ (Line: 547-564).

- Equation (10) needs an identity matrix to be correct: $C = (X'X + \sigma_\epsilon^2 / \sigma_\beta^2 I)^{-1} \sigma_\epsilon^2$

Answer: The equation is changed and now it is presented in Equation (7).

- Equation (11): what are \hat{u}_i and $\hat{\sigma}_i$? These are not defined anywhere. Please define these.

Answer: The equation has changed.

- You also state that windows where $\hat{\sigma}_i$ is uninformative are omitted are dropped (lines 598-605) – what does this mean? Just because you think the results are “too uncertain” doesn't mean that they are invalid. This must over-inflate your confidence in results and causes me to question the results presented. Dropping results that are “uninformative” is definitely not a Bayesian thing to do.

Answer: We have removed windows where the estimate of the standard deviation of the transition is larger than 10, which is relatively high because changes in temperature larger than 10 °C from land cover change is not realistic. This is a conservative approach to prevent excessive noise to the results, which have been ultimately compared against observations. This is explained in the text (Line: 578-579).

- Figure 6: If you have a *Bayesian* distribution, you would plot the distribution, not a bar with error bars. The bar plot of a mean with error bars is one of the worst plots – you have so many nice graphics and plots – why not use a box and whiskers plot or a violin plot instead? These are the common way of displaying Bayesian estimates because they convey that the estimate is a distribution, not a point estimate.

Answer: We have removed the references to Bayesian and focused on point estimates and associated uncertainty.

REVIEWERS' COMMENTS:

Reviewer #4 (Remarks to the Author):

I thank the authors for their efforts in revising the communication of the statistical model. I find the model much easier to understand now based on the revisions and have a better understanding of the results presented now that I can better understand the analyses.

The paper shows some weak evidence of a correlation between land cover class (LCC) and climate change using a local window penalized regression model. However, most of the results when averaged over space fail to reject the null hypothesis of no correlation between climate and LCC which makes the claims in the manuscript weaker than presented. No big deal, as this does not mean there is not evidence of a correlation – only that the effect averaged over space is not significantly different from 0.

For example, in lines 142-144 the authors claim the sign of the coefficient changes (relative to lines 118-120), but the 90% CI for both cases contains 0 and is mostly centered at 0 (well within the margin of error) suggesting that there is not enough evidence to support the claim of a pattern of change correlated with LCC when averaged across the domain. Figure 2 and Figure 6 attempt to resolve this issue by displaying the spatial variation in this coefficient but it is unclear if any of the spatially varying estimates are from statistically significant model estimates. I suspect there is a significant relationship due to the spatial coherence of the estimates – a plot of only the pixels with a significant non-zero regression line would help in the interpretation of the output.

There appear to be some significant changes in some categories (e.g., Figure 6, for some urban LCCs) but in general, most of these do not appear to be significantly different from 0 when averaged over the spatial domain. If the model allows for local spatial inference, why not just remove these spatially-averaged estimates as they are not really meaningful to the story? Figure 2 shows that there appears to be some weak evidence of a correlation between LCC and climate change – but there is not convincing evidence in this data for a causal relationship between LCC and climate. I do find claims like (in the abstract)

“At continental scale, the mean cooling is mainly driven by agriculture abandonment (cropland-to-forest transitions), but a novel approach based on ridge-regression decomposing the temperature change to the individual land transitions shows opposite responses to cropland losses and gains between western and eastern Europe.”

to not be supported purely based on the data because there is not evidence that the the climate change is “driven” by LCC – only that LCC and climate are correlated – and this correlation is weakly supported. The language should be changed to reflect that this interpretation is not causal – only correlation. It is entirely possible that the land cover changes are a direct result of changing climate where agricultural land use change is a function of the population reacting to a changing climate, the exact opposite causal result of what you claim. I am not an expert in the literature of land cover/climate relationship so if this is a well-known result in the field, this can be used to justify the causal relationship, but I don't find the data, on their own, to be indicative of this causal mechanism without other supporting theory (which I admit I am ignorant of).

The use of error bars of the mean +/- ?sd? (you call this a sd but is suspect it is an SE as the CIs are 1.645 times the claimed “sd” – this is the formula for a CI using the SE) in Figure 6 make it hard to visually assess the estimated effects from the model – revising this figure to plot the CIs (not bar plots plus error bars) would make this clear and not require me, the reader, to do a lot of work to understand and interpret the figure correctly. In the caption of figure 6 you say this is one standard deviation – I believe this is actually a standard error of the mean as your CIs' in the manuscript are approximately mean +/- 1.645 SEs (but you call these standard deviations) – this should be corrected. Again, this confusion over sd/SE is a common issue for barplots with error bars which makes interpreting these plots quite difficult and makes me, the reader, do the work to interpret the figure that you, the authors should be doing for me (and, I suspect, the miscommunicated interpretation of sd/SE also causes confusion).

Reviewer #4 (Remarks to the Author):

I thank the authors for their efforts in revising the communication of the statistical model. I find the model much easier to understand now based on the revisions and have a better understanding of the results presented now that I can better understand the analyses.

Thanks for the appreciation of our efforts.

The paper shows some weak evidence of a correlation between land cover class (LCC) and climate change using a local window penalized regression model. However, most of the results when averaged over space fail to reject the null hypothesis of no correlation between climate and LCC which makes the claims in the manuscript weaker than presented. No big deal, as this does not mean there is not evidence of a correlation – only that the effect averaged over space is not significantly different from 0. For example, in lines 142-144 the authors claim the sign of the coefficient changes (relative to lines 118-120), but the 90% CI for both cases contains 0 and is mostly centered at 0 (well within the margin of error) suggesting that there is not enough evidence to support the claim of a pattern of change correlated with LCC when averaged across the domain. Figure 2 and Figure 6 attempt to resolve this issue by displaying the spatial variation in this coefficient but it is unclear if any of the spatially varying estimates are from statistically significant model estimates. I suspect there is a significant relationship due to the spatial coherence of the estimates – a plot of only the pixels with a significant non-zero regression line would help in the interpretation of the output.

As explained in another comment below, we revised the paper and adapted the terminology to better reflect a correlational nature in the results, rather than a causality. The range is mostly driven by the spatial variability of the climate response, the seasonality, and the heterogeneous distribution of the land cover changes in each grid cell. For example, signals in summer are much stronger than the annual mean.

There appear to be some significant changes in some categories (e.g., Figure 6, for some urban LCCs) but in general, most of these do not appear to be significantly different from 0 when averaged over the spatial domain. If the model allows for local spatial inference, why not just remove these spatially-averaged estimates as they are not really meaningful to the story? Figure 2 shows that there appears to be some weak evidence of a correlation between LCC and climate change – but there is not convincing evidence in this data for a causal relationship between LCC and climate. I do find claims like (in the abstract) “At continental scale, the mean cooling is mainly driven by agriculture abandonment (cropland-to- forest transitions), but a novel approach based on ridge-regression decomposing the temperature change to the individual land transitions shows opposite responses to cropland losses and gains between western and eastern Europe.” to not be supported purely based on the data because there is not evidence that the the climate change is “driven” by LCC – only that LCC and climate are correlated – and this correlation is weakly supported. The language should be changed to reflect that this interpretation is not causal – only correlation. It is entirely possible that the land cover changes are a direct result of changing climate where agricultural land use change is a function of the population reacting to a changing climate, the exact opposite causal result of what you claim. I am not an expert in the literature of land cover/climate

relationship so if this is a well-known result in the field, this can be used to justify the causal relationship, but I don't find the data, on their own, to be indicative of this causal mechanism without other supporting theory (which I admit I am ignorant of).

We adapted the terminology in some statements, and highlighted the correlational nature of the results. However, the interpretation of the observed responses is generally based on background variables (shown in SI), existing theories, and are found to be consistent with other modelling and empirical studies. The existence of a causal relationship between LCCs and climate is consolidated in the scientific literature. A bunch of review studies offers good insights on the topic: Pielke et al., 2011; Mahmood et al., 2014; Perugini et al., 2017 (see manuscript for full references). The coupling between land cover changes and climate was also assessed by the latest IPCC Special Report on Climate Change and Land, Chapter 2, Section 2.5. In our experiments, the land cover is the only parameter that is changed relative to the control experiment, so any difference in temperature can be attributed to the change in land cover. As discussed in the paper, our results were broadly consistent with previous modelling studies and observational datasets.

The use of error bars of the mean \pm sd? (you call this a sd but I suspect it is an SE as the CIs are 1.645 times the claimed "sd" – this is the formula for a CI using the SE) in Figure 6 make it hard to visually assess the estimated effects from the model – revising this figure to plot the CIs (not bar plots plus error bars) would make this clear and not require me, the reader, to do a lot of work to understand and interpret the figure correctly. In the caption of figure 6 you say this is one standard deviation – I believe this is actually a standard error of the mean as your CIs' in the manuscript are approximately mean \pm 1.645 SEs (but you call these standard deviations) – this should be corrected. Again, this confusion over sd/SE is a common issue for barplots with error bars which makes interpreting these plots quite difficult and makes me, the reader, do the work to interpret the figure that you, the authors should be doing for me (and, I suspect, the mis-communicated interpretation of sd/SE also causes confusion).

We confirm that the error bars in Figure 6 refer to standard error of the mean. This is now corrected in the caption.